# Asymmetric sulfonamide design enabling high-voltage sodium-ion pouch cells in wide temperature

Xinke Cui[1], Qunfang Li[1], Gang Chang[2], Wei Tang[3], Xin Huang ®[4], Lang Huang ®[5], Xue Han[6] & Weijiang Xue ®[1] ✉

To advance toward commercial viability, sodium-ion batteries are required to operate under high cut-off voltages and low temperatures. This necessitates electrolyte designs that provide sufficient oxidative stability for high-voltage cycling while maintaining high ion mobility at low temperatures. Herein we design a non-flammable sulfonamide solvent molecule, N-ethyl-N-methyl-tri-fluoromethanesulfonamide, by introducing asymmetric alkyl substituents that create a geometric kink to hinder efficient crystal packing during cooling, leading to a low melting point (−86 °C). The resulting sulfonamide-based electrolyte exhibits weak ion–dipole interaction and favorable solvation structures enriched in contact ion pairs and aggregates, which promote the formation of highly stable and conductive interphases with both the positive and negative electrodes. Benefiting from these features, the sulfonamide-based electrolyte enables 1-Ah-level hard carbon‖NaNi$_{1/3}$Fe$_{1/3}$Mn$_{1/3}$O$_2$ pouch cells to retain 69.8% and 42.3% of room-temperature capacity even at −60 °C and −70 °C, while achieving capacity retentions of 90.0% and 81.6% after 1500 and 1000 cycles at high upper cut-off voltages of 4.15 V and 4.2 V, respectively. The sulfonamide-based electrolyte also improves the high-temperature cycling stability and delays the onset and trigger of thermal runaway at the pouch-cell level. This work offers fundamental insights into solvent molecule and electrolyte design for advancing high-energy and wide-temperature sodium-ion batteries.

Rechargeable sodium-ion batteries (SIBs) have attracted significant attention as promising candidates for energy storage applications due to the abundant Na resources and their potential low cost[1–5]. However, the practical deployment of SIBs is still hindered by their relatively low energy density[6–9], which compromises their competitiveness compared to commercial lithium-ion batteries (e.g., graphite‖LiFePO$_4$,

~180 Wh kg$^{-1}$)[10,11]. To boost their energy density, a promising approach is to elevate the charging voltage (e.g., ≥4.2 V vs. Na/Na$^+$, 4.2 V$_{Na}$) of the O3-type layered transition metal (TM) oxide positive electrodes (Na$_x$TMO$_2$, $0 < x \le 1$; TM: Ni, Co, Mn, Fe, etc.) to extract more Na[12–14]. However, this strategy often leads to interphasial instability and poor cycling performance due to severe electrolyte oxidation at high

[1]State Key Laboratory for Mechanical Behavior of Materials, Center for Advancing Materials Performance from the Nanoscale (CAMP-Nano), Xi'an Jiaotong University, Xi'an, China. [2]Instrumental Analysis Center of Xi'an Jiaotong University, Xi'an, China. [3]School of Chemical Engineering and Technology and National Innovation Platform (Center) for Industry-Education Integration of Energy Storage Technology, Xi'an Jiaotong University, Xi'an, China. [4]School of Chemistry and Chemical Engineering, Shandong University of Technology, Zibo, China. [5]Qingdao Industrial Energy Storage Research Institute, Qingdao Institute of Bioenergy and Bioprocess Technology, Chinese Academy of Sciences, Qingdao, China. [6]State Key Laboratory of Metal Matrix Composites, Shanghai Jiao Tong University, Shanghai, China. ✉e-mail: xuewj@xjtu.edu.cn

potentials[15]. Moreover, although the desolvation of $Na^+$ at low temperatures is theoretically easier than that of $Li^+$—due to its ~30% larger ionic radii[16,17] and thus weaker electrostatic interactions with solvent molecules—the low temperature performance of SIBs remains unsatisfactory, especially under low temperature conditions such as −60 °C or below[18,19]. These challenges under high voltages and extreme temperature conditions highlight the critical need for electrolyte designs that can simultaneously withstand high voltages and enable efficient ion transport at low temperatures.

Unlike metal negative electrodes[20–22], where $Na^0$ (Na-metal) electrodeposition typically occurs in a surface-confined and planar manner with minimal ion diffusion, hard carbon (HC) negative electrodes rely on intercalation chemistry[23,24] that requires $Na^+$ cations to diffuse across a several-tens-μm-thick electrode through tortuous internal pores. This longer and more complex transport pathway makes efficient $Na^+$ intercalation/deintercalation particularly challenging under practical conditions (assuming a typical HC areal loading of 5 mg cm$^{-2}$ and a compressed density of 1 g cm$^{-3}$, which corresponds to a ~ 50 μm electrode thickness), especially at low temperatures. In this scenario, electrolyte design should particularly focus on facilitating $Na^+$ diffusion and weakening ion–dipole interactions, while also ensuring high-voltage stability and sufficient ionic conductivity. Recent work in this field has relied on advances in introducing effective additives[25,26] into conventional carbonate-based electrolytes, as well as developing efficient ether-based[27–29] and acetonitrile-based[30] electrolytes. While these strategies have led to great progress, it is still challenging to simultaneously meet all the above mentioned requirements—particularly when scaling up to ampere-hour (Ah)-level pouch cells[13]. Further progress requires molecular design to simultaneously address the demands of high-voltage stability, low temperature performance, and practical scalability.

Ion–dipole interactions[31,32] play an important role in desolvation, which dictates the kinetics. From a fundamental perspective, the $Na^+$ cation can be treated as a point charge. As illustrated in Fig. 1a, consider a $Na^+$ cation located at a distance $r$, from the center of a polar molecule with dipole moment $\mu$, forming an angle $\theta$ relative to the dipole axis. The maximum attractive interaction (i.e., maximum negative energy) occurs when the dipole is aligned directly away from the ion ($\theta = 0°$). In realistic electrolyte environments, thermal fluctuations lead to continuous reorientation of dipolar solvent molecules with a rotation angle of $\beta$ (around $\theta = 0°$). To account for this rotational averaging, the ion–dipole interaction energy $\omega$ is expressed in an integral form over all possible orientations:

$$\omega = -\int_0^\beta \frac{e\mu\cos\theta}{4\pi\varepsilon_0\varepsilon r^2}d\theta \qquad (1)$$

Where $\varepsilon$ and $\varepsilon_0$ are dielectric permittivities of the dipole and vacuum[33]. As temperature decreases, the rotational degrees of dipolar solvent molecules are progressively restricted, leading to reduced molecular reorientation ($\beta\downarrow$). This effect becomes particularly pronounced as the temperature approaches the solvent's melting point ($T_m$), where molecular (i.e., dipolar) rotations effectively freeze (Fig. 1b). The dielectric constant of the solvent ($\varepsilon$) drops sharply upon solidification, reflecting the diminished ability of the medium to reorient dipoles in response. According to Eq. (1), it leads to much higher ion–dipole interaction at low temperature than at room temperature (25 °C, RT) ($\omega_{LT}\gg\omega_{RT}$). Therefore, the molecular design of solvents with $T_m$ far below the intended operating temperature is critical to ensure sufficient dipolar mobility and maintain a high dielectric constant for effective ion transport.

To design solvent molecules with intrinsically low $T_m$, one must consider not only the strength of intermolecular attractive forces but also the molecular geometry that governs crystal packing. While

boiling points are largely dictated by attractive interactions, $T_m$ are primarily influenced by how well molecules can pack into an ordered solid lattice. According to this principle, molecules with rigid, symmetrical structures tend to pack efficiently and thus exhibit high $T_m$. In contrast, introducing asymmetry or steric irregularities disrupts lattice formation and lowers $T_m$. Based on this rationale, a non-flammable sulfonamide molecule, N-ethyl-N-methyl-trifluoromethanesulfonamide (EMTMSA), was rationally designed by incorporating asymmetric alkyl substituents (methyl and ethyl groups) on the sulfonamide nitrogen (Fig. 1c), which introduce a geometric kink and hinder efficient crystalline packing. This molecular asymmetry disrupts solid-state packing, resulting in poor crystallinity and a low melting point of −86 °C (Fig. 1d). In addition, its high flash and boiling points, along with moderate viscosity and dielectric constant, render it a well-balanced and practically viable electrolyte solvent (Supplementary Table 1). Moreover, it also successfully inherits the anodic stability of its structural analog, N,N-dimethyltrifluoromethanesulfonamide (DMTMSA, Fig. 1c), which was previously developed by our group for high-voltage chemistries[34,35]. Specifically, as expected, the formulated EMTMSA-based electrolyte shows sufficient molecular rotational mobility at low temperatures, as revealed by the longer spin-spin relaxation time in variable-temperature nuclear magnetic resonance (VT-NMR). The electrolyte also exhibits large portions of contact-ion pairs (CIPs) and ion–solvent–anion aggregates (AGGs) with preferential solvation characteristics, promoting the formation of solid–electrolyte interphase (SEI) with low resistance and stability, effectively supporting low temperature and long-term cycling stability at high voltages. Benefiting from these, our EMTMSA-based electrolyte successfully enables an Ah-level HC∥NaNi$_{1/3}$Fe$_{1/3}$Mn$_{1/3}$O$_2$ (NFM) pouch cell to maintain 69.8% and 42.3% of its room-temperature capacity even at −60 and −70 °C, respectively, while simultaneously delivering high temperature stability and delayed thermal runaway. The pouch cells also exhibit long-term cycling stability with high capacity retentions (with respect to (w.r.t.) the third cycle) of 90.0% (cut-off voltage ~4.15 V$_{Na}$) and 81.6% (cut-off voltage ~4.2 V$_{Na}$) after 1500 and 1000 cycles, respectively. This work provides valuable insights for designing electrolyte molecules or recipes for versatile and practical SIBs.

## Results
### Molecular and electrolyte design
While polar molecules are beneficial for dissociating sodium salts to support high ionic conductivity, they often exhibit higher $T_m$ than their nonpolar counterparts. This is primarily due to stronger intermolecular Coulombic interactions—such as dipole–dipole interactions or hydrogen bonding—which enhance molecular ordering and raise the crystallization temperature. To elucidate the nature of intermolecular interactions, molecular dynamics (MD) simulations were performed for three sulfonamides with different alkyl substitutions: DMTMSA, EMTMSA, and DETMSA (N,N-diethyl-trifluoromethanesulfonamide) at different temperatures (Supplementary Fig. 1 and Supplementary Data 1). In Fig. 1e, the Coulomb and van der Waals forces of DMTMSA are set as the reference (zero point), indicated by the dashed line at the center. The relative differences in interaction forces for EMTMSA and DETMSA are plotted accordingly. Substituting the two methyl groups in DMTMSA with longer or bulkier alkyl chains—either one ethyl (EMTMSA) or two ethyls (DETMSA)—results in enhanced van der Waals interactions. This increase is attributed to the larger molecular size and greater intermolecular contact area, which promote stronger dispersion forces. Conversely, the Coulombic interactions become weaker with increasing alkyl chain length, likely due to slightly reduced molecular polarity and dielectric response as more nonpolar alkyl groups are introduced. Importantly, the combined intermolecular force (Coulomb + van der Waals) is lower for both DETMSA and EMTMSA compared to DMTMSA, suggesting a

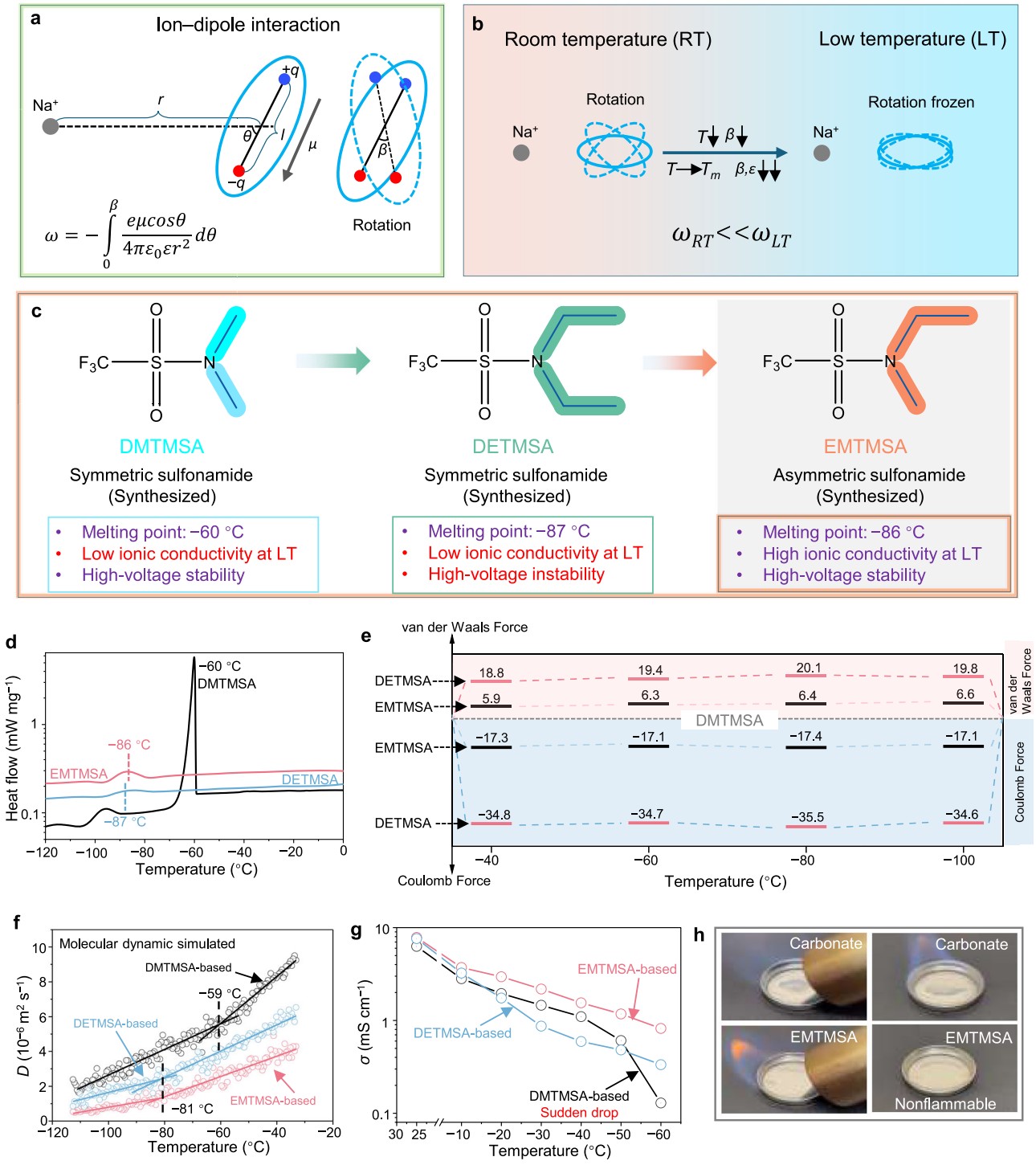

weaker overall interaction. This is consistent with both the experimentally measured $T_m$ and MD-simulated diffusion behavior. Differential scanning calorimetry (DSC) shows that EMTMSA and DETMSA exhibit no sharp endothermic peaks above −85 °C, while DMTMSA displays a distinct melting transition at −60 °C (Fig. 1d). MD-simulated diffusion coefficients exhibit a clear transition in slope around their $T_m$ with molecular mobility increasing more rapidly above the phase transition (Fig. 1f).

To formulate a practical electrolyte suitable for pouch cells, propylene carbonate (PC, $T_m = −49$ °C) and ethyl methyl carbonate (EMC, $T_m = −53$ °C)[36] were introduced as co-solvents with EMTMSA. The resulting sulfonamide-based electrolyte consists of 1 M sodium bis(fluorosulfonyl)imide (NaFSI) dissolved in an EMTMSA/PC/EMC

mixture with a volume ratio of 6:1:3. DMTMSA- and DETMSA-based electrolytes were prepared using the same recipe, with EMTMSA simply replaced by DMTMSA or DETMSA, respectively. In this recipe, 10 vol% PC primarily facilitates the dissociation of NaFSI due to its high dielectric constant, while 30 vol% EMC serves to reduce the overall viscosity of the electrolyte and enhance ion mobility. All solvents were subjected to molecular-sieve drying to eliminate residual water and consequently suppress possible hydrogen fluoride (HF) generation (Supplementary Fig. 2), which is critical for maintaining stable electrochemical performance. Correspondingly, a carbonate electrolyte consisting of 1 M NaFSI in PC/EMC (3:7 vol) was selected as the reference electrolyte. The DMTMSA- and EMTMSA-based electrolytes maintain high oxidative stability up to ≈4.8 V$_{Na}$, whereas the DETMSA-

**Fig. 1 | Molecular design for an electrolyte for high-voltage SIBs under low temperature conditions. a** Schematic of ion–dipole interaction energy $\omega$ between a unit charge $e$ and a dipole moment $\mu$ oriented at different angles $\theta$ relative to the dipole axis. Thermal fluctuations lead to continuous reorientation of dipolar solvent molecules with a rotation angle of $\beta$. **b** As temperature decreases, molecular rotation becomes increasingly restricted ($\beta\downarrow$) and this effect becomes especially pronounced near the solvent's melting point ($T_m$), where molecular rotation is effectively frozen ($\beta\downarrow\downarrow$). As a result, the dielectric permittivity $\varepsilon$ drops sharply upon approaching $T_m$, leading to a significant increase in the ion–dipole interaction energy $\omega$, which in turn hampers ion desolvation and transport. **c** Molecular design logic for the sulfonamide analogs, DMTMSA, DETMSA, and EMTMSA. Boxed areas indicate key physicochemical properties such as $T_m$, ionic conductivity, and anodic stability. **d** Differential scanning calorimetry (DSC) curves of DMTMSA, DETMSA, and EMTMSA, showing their thermal behaviors upon cooling. Both DETMSA and EMTMSA exhibit lower $T_m$ (−86 to −87 °C) compared to DMTMSA (−60 °C). **e** Van der Waals and Coulomb forces were calculated by molecular dynamics (MD) simulations of the three sulfonamides at different low temperatures. The black bars represent the differences in interaction forces at each temperature point, with DMTMSA serving as the reference baseline (zero). The unit of the forces is kJ mol⁻¹. **f** MD-simulated diffusion coefficients of the three sulfonamides as a function of temperature to determine their $T_m$. **g** Ionic conductivities ($\sigma$) of the DMTMSA-, EMTMSA-, and DETMSA-based electrolytes at different temperatures. Symmetric steel‖steel cells were assembled using different electrolytes, and electrochemical impedance spectroscopy (EIS) measurements were conducted at various temperatures to determine the corresponding ionic conductivities. **h** Ignition tests comparing the EMTMSA-based electrolyte with the conventional carbonate-based electrolyte. The EMTMSA-based electrolyte demonstrates largely improved flame retardancy. Source data are provided as a Source data file.

based electrolyte exhibits a greatly reduced anodic stability (Supplementary Fig. 3a). This decrease correlates with the substituent's electronic structure: longer alkyl chains donate more electron density, weakening the $FSO_2-$ group's electron-withdrawing effect and elevating the local electron density, consistent with the density functional theory (DFT) results showing higher highest occupied molecular orbital (HOMO) levels for longer chains (Supplementary Fig. 3b and Supplementary Data 2). The ionic conductivities of all three sulfonamide-based electrolytes decrease with decreasing temperature, but the EMTMSA-based electrolyte consistently maintains the highest conductivity. Notably, the DMTMSA-based electrolyte exhibits a sudden drop in conductivity at −60 °C, indicating a possible crystallization around this temperature (Fig. 1g). Furthermore, we performed ignition tests by exposing both the carbonate-based and EMTMSA-based electrolytes to an open flame (Fig. 1h, Supplementary Movies 1 and 2). Compared to the carbonate reference electrolyte, the EMTMSA-based electrolyte demonstrates largely improved flame retardancy, effectively suppressing ignition and sustaining nonflammability, which is critical for improving battery safety. Moreover, the EMTMSA-based electrolyte exhibits much better wettability than the reference carbonate electrolyte (Supplementary Fig. 4). Therefore, considering the ionic conductivities at low temperatures and electrochemical stability windows of the sulfonamide-based electrolytes, the designed EMTMSA-based electrolyte with an asymmetric molecular characteristic was selected as the optimal formulation.

## The coordination interactions and solvation structures in the electrolytes

The characteristics of the $Na^+$ solvation structures play an important role in influencing the decomposition of electrolyte components on both the positive electrode and negative electrode surfaces. The S–N–S symmetric stretching vibrational mode for the neat NaFSI salt is around 770–780 cm⁻¹ (Supplementary Fig. 5), which undergoes a red shift when dissolved in solvents, driven by the Coulombic interactions. Raman spectra of both the EMTMSA- and carbonate-based electrolytes were deconvoluted to identify the various ion-solvent clusters, including solvent-separated ion pairs (SSIP, ~720 cm⁻¹), CIPs (~735 cm⁻¹), and ion-pair aggregates (AGG, ~745 cm⁻¹)[37]. While SSIP (56.4%) is the dominant cluster type in the carbonate-based electrolyte (Fig. 2a), AGG (10.2%) and CIP (84.2%) are the main species in the EMTMSA-based electrolyte (Fig. 2b), indicating that more $FSI^-$ anions participate in the first solvation sheath. It would promote increased anion decomposition near the charged electrode surfaces, which is beneficial for enhancing the electrochemical stability of the electrolyte. To elucidate the exact coordination sites between ions and solvents, an NMR technique was conducted on the electrolytes before and after incorporating NaFSI. An apparent upfield shift of 9.22 ppm for PC was observed from the ¹⁷O NMR in the carbonate-based electrolyte on dissolving 1 M NaFSI (Fig. 2c), indicating the preferential

solvation of $Na^+$ by PC molecules, attributed to the much higher dielectric constant of PC than that of EMC. When using EMTMSA/PC/EMC as the solvent system, a similar preferential solvation phenomenon was observed (Fig. 2d, Supplementary Fig. 6). An upfield shift of 1.01 ppm for the O atoms in the EMTMSA molecule was also detected upon dissolving 1 M NaFSI (Fig. 2d), suggesting that EMTMSA participates in the $Na^+$ solvation sheath. The 1.01 ppm shift is appreciable considering EMTMSA constitutes the largest volume fraction (60%) among the mixed solvents. A downfield shift in ¹⁹F NMR can also be noted from the EMTMSA molecule after salt incorporation (Supplementary Fig. 7). It should be noted that NMR spectroscopy reflects an ensemble-averaged chemical environment, which supports the conclusion that EMTMSA contributes to the $Na^+$ solvation structure. Based on the above arguments, schematic figures of the representative $Na^+$ solvation structures in both electrolytes were presented in Fig. 2e, f.

Besides the characterizations of the solvation structures at 25 °C, VT-NMR was also employed to reveal the chemical environment and molecule mobility of the electrolytes at low temperatures[38]. When the temperature decreases from RT to −50 °C, upfield shifts were noted for all ¹H NMR spectra (Fig. 2g, Supplementary Fig. 8). This could be attributed to the reduced thermal motion of the electrons at lower temperatures, leading to increased electron density around each nucleus and enhanced shielding effects. More importantly, noticeable peak broadening was observed for all signals in the ¹H NMR spectra of the PC/EMC solvent mixture (Fig. 2g), suggesting slower molecular dynamics and increased viscosity at lower temperatures. Higher viscosity leads to a shorter spin-spin relaxation time ($T_2$), which is inversely proportional to the NMR line width. In stark contrast, the peaks in the EMTMSA/PC/EMC solvent mixture remain sharp and almost unchanged even at −50 °C (Fig. 2h), indicating preserved molecular mobility and lower viscosity. Notably, the ¹H NMR signals of pure EMTMSA exhibit minimal changes during cooling (Supplementary Fig. 9), suggesting that EMTMSA itself can maintain stable molecular dynamics upon cooling. This highlights the positive effect of EMTMSA, which helps maintain a fluidic environment and prevents molecule rotational motions from slowing, validating our rationale for designing the asymmetric EMTMSA molecule with an intrinsically low $T_m$.

## Electrochemical performance of SIB pouch cells at room and low/high temperatures

To evaluate the electrolyte performance under practical cell conditions, Ah-level pouch cells were assembled by multilayer-stacking NFM positive electrodes (areal loading 40 mg cm⁻², double-sided) and HC negative electrodes (areal loading 20 mg cm⁻², double-side, detailed cell parameters are provided in Supplementary Table 2). It is worth noting that the mass transport characteristics and failure mechanisms in pouch cells with high-loading electrodes differ substantially from those in coin cells. The more stringent conditions in pouch cells demand adequate electrolyte wettability to ensure uniform infiltration

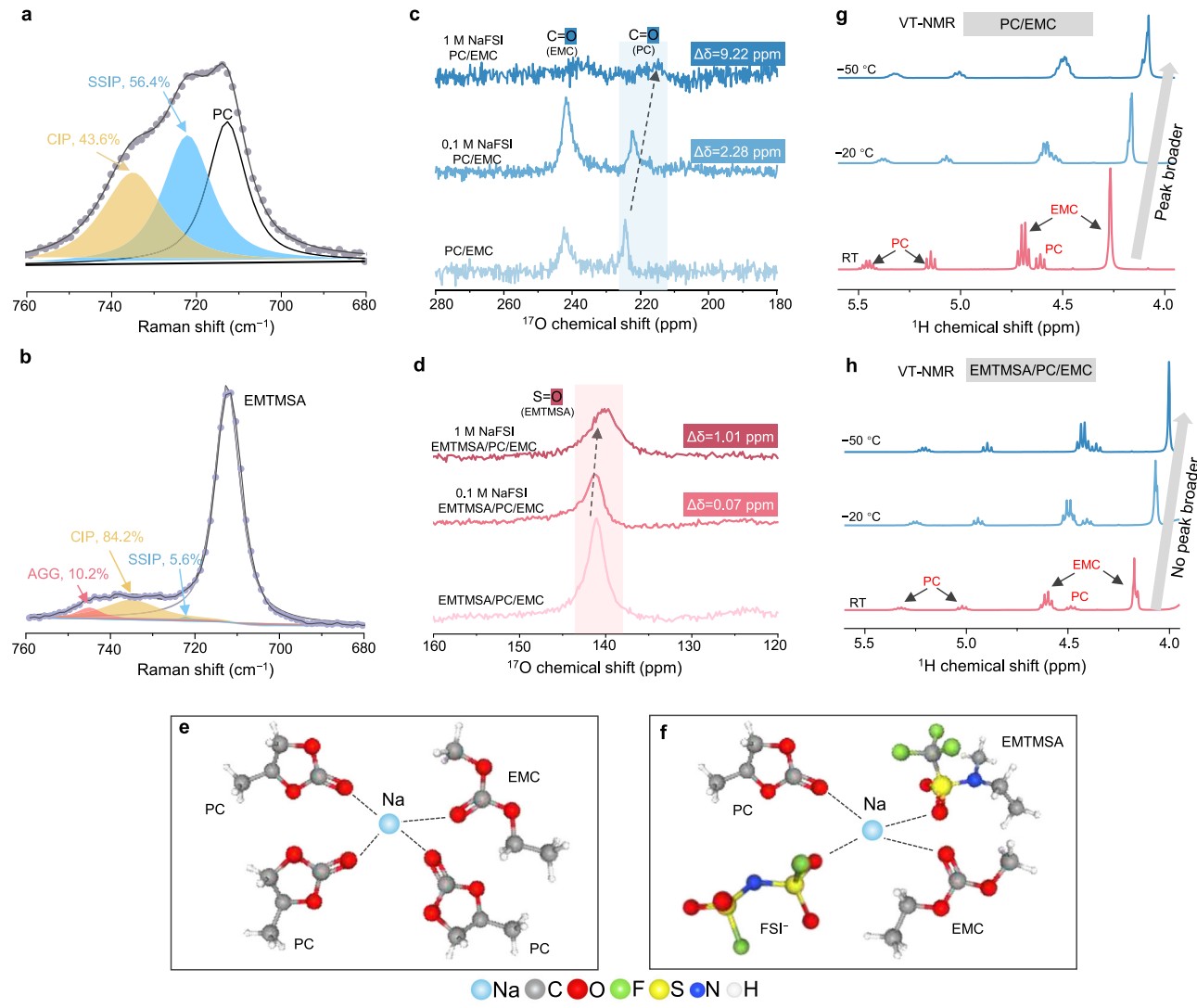

**Fig. 2 | The coordination interactions between Na⁺ and different solvent molecules in the EMTMSA-based and carbonate-based electrolytes by Raman spectroscopy and NMR techniques.** Deconvolution of Na⁺–FSI⁻ Raman peaks for the carbonate-based (**a**) and EMTMSA-based (**b**) electrolytes with peak fits for different cluster types, including SSIP, CIP, and AGG. ¹⁷O NMR spectra of the carbonate-based (**c**) and EMTMSA-based (**d**) electrolytes before and after incorporating 0.1 M and 1 M NaFSI into the solvent mixtures. The corresponding functional groups in different molecules and chemical shifts are denoted. Schematic figures of the representative Na⁺ solvation structures in the carbonate-based (**e**) and EMTMSA-based (**f**) electrolytes. ¹H VT-NMR spectra of the PC/EMC (**g**) and EMTMSA/PC/EMC (**h**) solvent mixtures at 25 °C, −20 °C, and −50 °C. Peak broadening is observed for the carbonate mixtures upon cooling, corresponding to significantly decreased molecular rotation. In contrast, the peaks remain almost unchanged for the EMTMSA-carbonate mixture. Source data are provided as a Source data file.

across the entire electrode stack. Furthermore, gas evolution within pouch cells can induce rapid capacity decay due to cell swelling and interfacial disconnection that are negligible in coin cell configurations. The gas generated during the formation cycle was completely removed by a vacuum degassing process during the final sealing step after formation (Supplementary Fig. 10). A small gas pocket remains after final sealing, serving as a visible indicator for qualitative observation of gas evolution during long-term cycling, after which the pouch cells are placed in a mechanical fixture for cycling tests (Supplementary Fig. 10). The long-term cycling performance of pouch cells using the carbonate-based and EMTMSA-based electrolytes was first evaluated with an upper charging voltage of 4.1 V at a charge–discharge rate of 1 C (4.15 V_Na, Fig. 3a). Upon desodiation to 4.15 V_Na, a short voltage plateau was observed (Supplementary Fig. 11), which corresponds to the detrimental P3 to OP2 phase transformation[13,39]. Such a transformation evokes appreciable

instability within the bulk lattice and the interphasial region, eventually deteriorating cycling stability. To avoid such degradation and achieve a long-term cycling stability (e.g., >1000 cycles), it is generally necessary to restrict the upper cut-off voltage to ≤4.0 V_Na. However, even under an elevated upper cut-off voltage of 4.15 V_Na, our EMTMSA-based electrolyte can enable cycling stability with 90.0% capacity retention (w.r.t. the third cycle) after 1500 cycles (Fig. 3a). It outperforms the carbonate-based electrolyte, which showed only 72.0% capacity retention (w.r.t. the third cycle) under the same conditions (Fig. 3a). More importantly, when evaluated at a higher cut-off voltage of 4.2 V_Na (Fig. 3b), the pouch cell with our EMTMSA-based electrolyte can still retain 81.6% of its initial capacity (w.r.t. the third cycle) after 1000 cycles at a charging-discharging rate of 0.5 C (close to the capacity retention 81.9% achieved by the pouch cell with the DMTMSA-based electrolyte, Supplementary Fig. 12), outperforming the carbonate-based and DETMSA-based electrolytes, which retain 65.6%

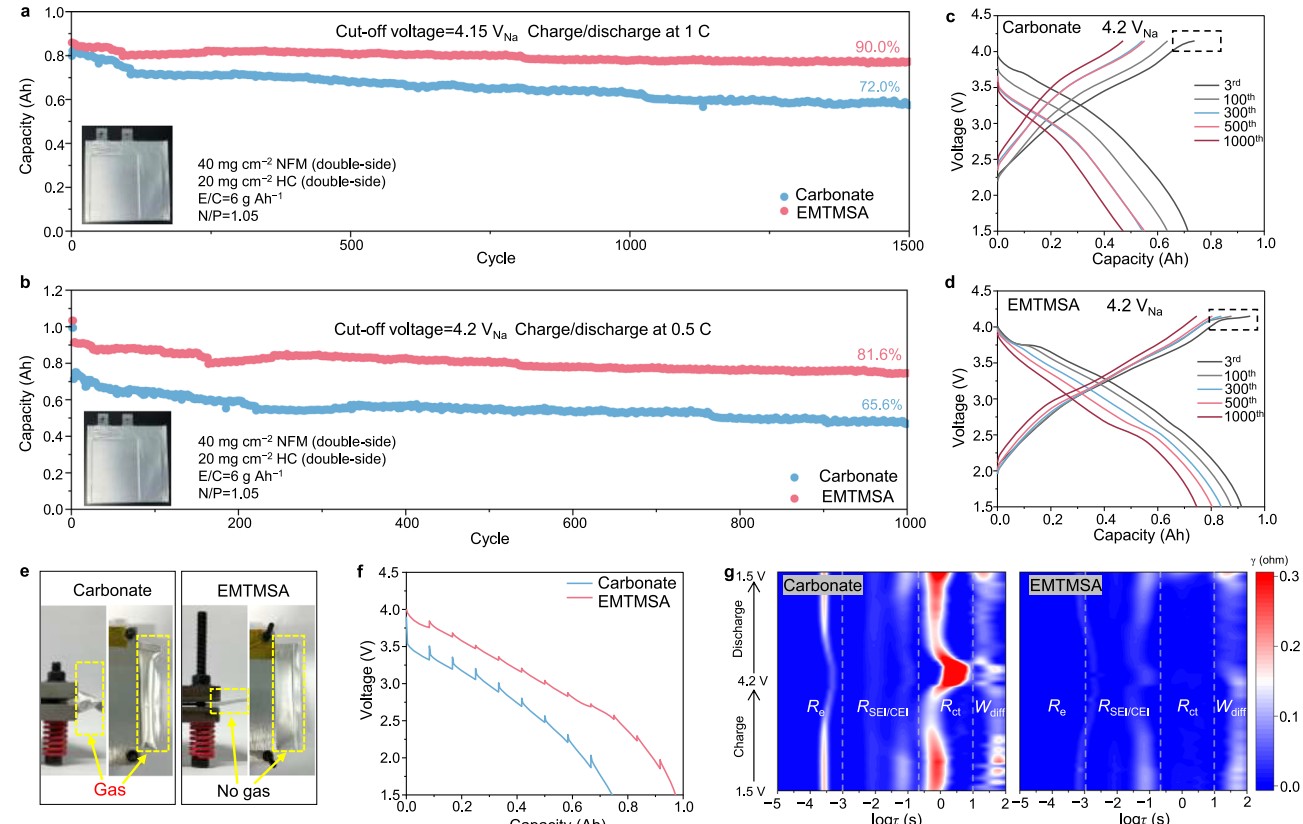

**Fig. 3 | Electrochemical performance of the HC∥NFM pouch cells with different electrolytes at RT.** Long-term cycling performance of the pouch cells with upper cut-off voltages of 4.15 $V_{Na}$ (**a**) and 4.2 $V_{Na}$ (**b**). Voltage profiles corresponded to (**b**) with the carbonate-based (**c**) and EMTMSA-based (**d**) electrolytes at 4.2 $V_{Na}$ cut-off voltage. The electrolyte-to-capacity ratio (E/C) and negative-to-positive capacity ratio (N/P) of the pouch cells are indicated in the figure. The pouch cells have 8 positive electrode layers and 9 negative electrode layers. All electrochemical tests on pouch cells were conducted at a 1 A defined as 1 C rate. **e** Photographs of pouch cells after 1000 cycles with different electrolytes, highlighting distinct gassing behaviors. The cell with the carbonate-based electrolyte exhibits severe gas accumulation, as evidenced by the visibly swollen pouch. In contrast, it is largely suppressed in our EMTMSA-based electrolyte. **f** Discharge voltage profiles of galvanostatic intermittent titration technique measurements on the pouch cells after 200 cycles in both electrolytes. **g** In situ distribution of relaxation times data of the pouch cells with carbonate-based (left) and EMTMSA-based (right) electrolytes at 25 °C. Blue and red colors correspond to the minimum and maximum intensities, respectively. Source data are provided as a Source data file.

and 64.5% of its original capacity for 1000 cycles under the same conditions, respectively.

The elevation in the charging voltage from 4.0 to 4.2 $V_{Na}$ enables a ~25% increase in the specific capacity of the NFM positive electrode (Supplementary Fig. 13), attributed to the extended voltage plateau near 4.15 V during the charging process (Fig. 3c, d, indicated by dashed boxes). These results demonstrate that our EMTMSA-based electrolyte can effectively passivate the aggressive NFM surfaces by forming a highly conductive and stable cathode–electrolyte interphase (CEI). In stark contrast, the carbonate-based electrolyte gives rise to an unstable and resistive CEI, as evidenced by its much lower discharge capacity of ~0.71 Ah at 0.5 C, compared to that of the cell with the EMTMSA-based electrolyte (~0.91 Ah, Fig. 3b). Compared to the carbonate-based electrolyte, the pouch cell with the EMTMSA-based electrolyte also delivers improved average Coulombic efficiency (CE) at 4.2 $V_{Na}$ (~99.94%, Supplementary Fig. 14) and better rate performance (Supplementary Fig. 15). The formation of an unstable CEI in the carbonate-based electrolyte under high-voltage cycling is further evidenced by the pronounced gas evolution, as indicated by the noticeable swelling of the gas pocket (Fig. 3e). In contrast, negligible gas generation was observed in the cell employing the EMTMSA-based electrolyte (Fig. 3e), highlighting its enhanced interphasial stability. To analyze the impedance evolution during cycling, galvanostatic intermittent titration technique (GITT) measurements were performed on

the pouch cells after 200 cycles in different electrolytes. The overpotentials observed in the cell with the EMTMSA-based electrolyte were significantly lower than those in the cell with the carbonate-based electrolyte (Fig. 3f), further verifying the different interphasial stability in both electrolytes. Moreover, the evolution of desodiation and sodiation kinetics was investigated using in-situ electrochemical impedance spectroscopy (EIS), and the distribution of relaxation times (DRT) technique was employed to decouple electrochemical processes over different timescales during the charge–discharge process[40,41]. The four distinct regions in DRT data (Fig. 3g) are attributed to $R_e$ (electrolyte resistance), $R_{SEI/CEI}$ (SEI and CEI resistance), $R_{ct}$ (charge transfer resistance), and $W_{diff}$ (solid-state diffusion). During the charge–discharge process, the cell with the carbonate-based electrolyte exhibits significant fluctuations in $R_{ct}$, with high values especially near the high-voltage region (~4.2 V). In contrast, the $R_{ct}$ is almost negligible in the EMTMSA-based electrolyte, which further elucidates the low impedance of pouch cells with the EMTMSA-based electrolyte after high-voltage cycling. Additionally, the EMTMSA-based electrolyte also shows a smaller $R_{SEI/CEI}$ compared to the carbonate-based electrolyte, suggesting the formation of a more robust CEI/SEI and more efficient interphasial charge transfer. These results can also be verified by the impedance analysis on the pouch cells before and after long-term cycling (Supplementary Fig. 16 and Supplementary Table 3).

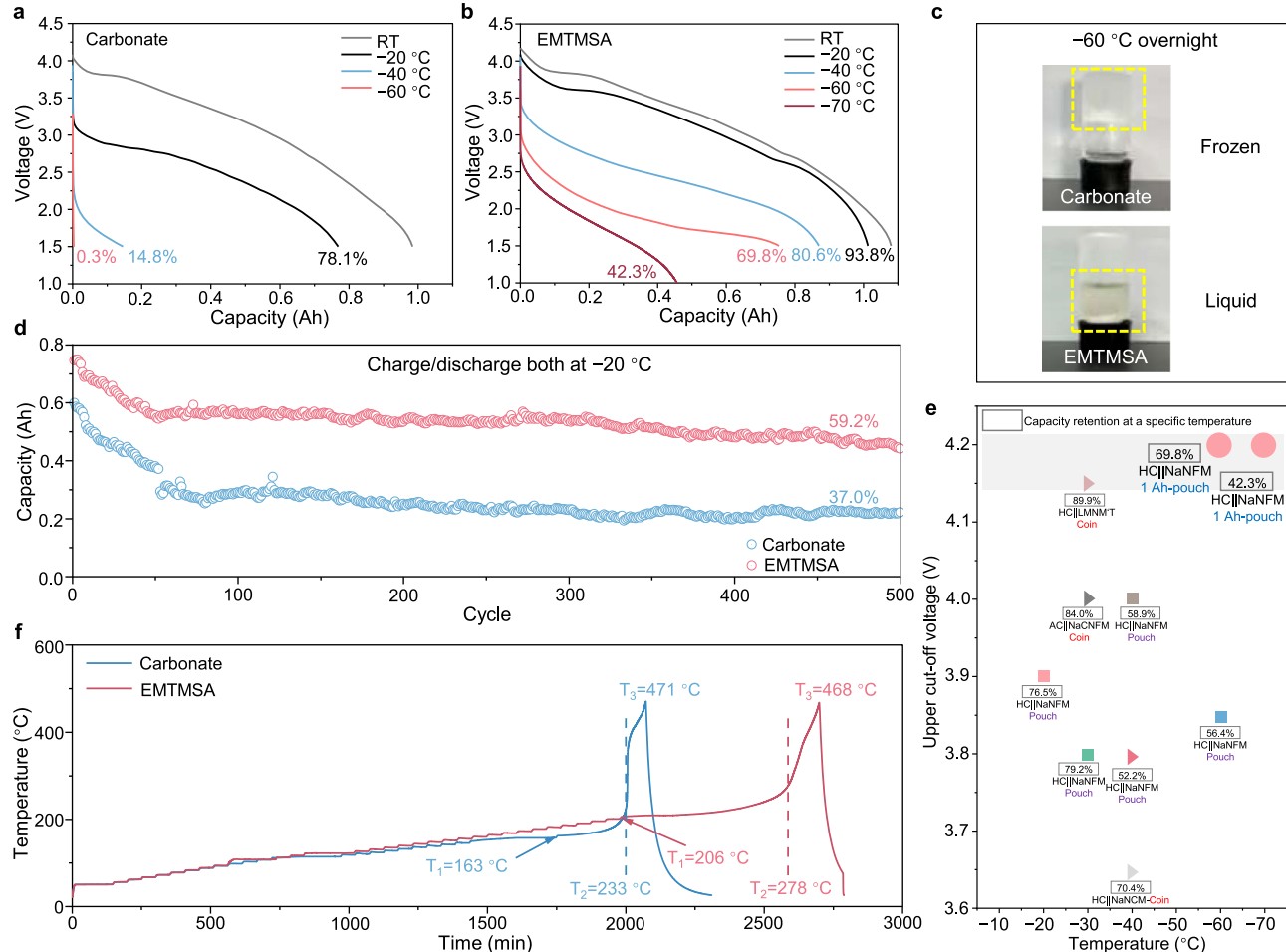

**Fig. 4 | Electrochemical performance of HC∥NFM pouch cells with different electrolytes at low/high temperature conditions.** Voltage profiles of pouch cells with the carbonate-based (**a**) and EMTMSA-based (**b**) electrolytes at 25 °C, −20 °C, −40 °C, −60 °C, and −70 °C. Discharge tests were performed at 0.1 C for all temperatures except −70 °C, where a rate of 0.01 C was used. **c** Optical photographs of different electrolytes after overnight storage at −60 °C. The carbonate-based electrolyte was frozen while the EMTMSA-based electrolyte remained liquid. **d** Cycling performance of pouch cells with both charging and discharging at −20 °C and 0.1 C. All electrochemical tests on pouch cells were conducted at a 1 A defined as 1 C rate. **e** Comparison of electrochemical performance between previous reports and this work, based on upper cut-off voltages and capacity retentions at a specific low temperature. The cell with the EMTMSA-based electrolyte in this study enables notable high-voltage operation and capacity retention at low temperature, outperforming some recent arts. **f** Temperature evolution monitored during the thermal runaway of fully charged pouch cells under ARC testing. Where $T_1$ represents the self-generated heat initiation temperature, $T_2$ represents the thermal runaway trigger temperature, and $T_3$ represents the maximum thermal runaway temperature. Compared to pouch cells with the carbonate-based electrolyte, these three key indicators are all reduced in pouch cells with the EMTMSA-based electrolyte. Source data are provided as a Source data file.

Applications such as grid-scale energy storage and electric tools in high-latitude countries demand that batteries operate reliably across a wide temperature range (−60 to 60 °C)[42]. Under low temperatures, it is crucial for SIBs to retain appreciable discharge capacity. This challenge is further amplified in pouch cells with high-loading electrodes, where low temperatures not only limit intercalation kinetics but also severely hinder ion transport, making it difficult to sustain sufficient ionic mobility throughout the electrode stack. Ah-level pouch cells were also used to evaluate the practical electrochemical performance at low temperatures (−20 °C, −40 °C, −60 °C, and −70 °C) at an upper charging voltage of 4.2 $V_{Na}$ with the electrolytes. While the pouch cell using the carbonate-based electrolyte retains 78.1% of its RT capacity at −20 °C, its performance deteriorates sharply with further temperature reduction by retaining only 14.8% at −40 °C and dropping to nearly zero at −60 °C (Fig. 4a). Similarly, the capacity of the pouch cell using the DMTMSA-based electrolyte declines to nearly zero at −60 °C (Supplementary Fig. 17). In sharp contrast, the pouch cell with the EMTMSA-based electrolyte exhibits much better low-temperature performance, retaining 93.8%, 80.6%, 69.8%, and 42.3% of its RT

capacity at −20, −40, −60, and −70 °C, respectively (Fig. 4b). The pouch cell with the EMTMSA-based electrolyte maintained high capacity retention, even under simultaneous low-temperature charging and discharging at 0.01 C, retaining 81.1%, 59.7%, and 41.9% of its RT capacity at −20, −40, and −60 °C, respectively (Supplementary Fig. 18).

This stark difference in low temperature electrochemical performance can be attributed to the distinct physical appearances of the two electrolytes after overnight storage at −60 °C: the carbonate-based electrolyte had solidified, while the EMTMSA-based electrolyte remained in a liquid state (Fig. 4c). Benefiting from its intrinsically low $T_m$ (−86 °C), EMTMSA constitutes 60 vol% of the EMTMSA-based electrolyte and serves as a fluidic medium that accommodates both PC and EMC molecules as well as solvated Na⁺ ions. This enables the electrolyte to maintain a liquid state even at −60 °C, ensuring continuous ion transport under low temperature conditions. Furthermore, we also evaluated the cycling performance of the pouch cells using different electrolytes, where both charging and discharging were performed at −20 °C. This scenario presents greater challenges to cycling stability, as the increased overpotential at low temperatures

can readily drive the negative electrode potential well below 0 $V_{Na}$. Such conditions promote the irreversible $Na^0$ plating on the HC surface, leading to the loss of Na inventory and thus rapid capacity fading. Compared to the carbonate-based electrolyte, the pouch cell with the EMTMSA-based electrolyte delivers improved cycling stability and CE at −20 °C (Fig. 4d; Supplementary Fig. 19). After 500 cycles at −20 °C, the cell using the EMTMSA-based electrolyte retained 59.2% of its initial capacity in contrast to only 37.0% retention observed with the carbonate reference electrolyte. In comparison with recent studies on low-temperature electrolytes for SIBs with layered oxide positive electrodes, our results with the EMTMSA-based electrolyte demonstrate notable high-voltage (4.2 $V_{Na}$) and capacity retention under low temperature (−60 °C and −70 °C) conditions (Fig. 4e and Supplementary Table 4). Notably, these results were achieved in practical pouch cells. Moreover, the high temperature stability at 45 °C and 60 °C was evaluated in pouch cells at an upper charging voltage of 4.2 $V_{Na}$. Pouch cells with the EMTMSA-based electrolyte exhibit better cycling stability compared with those employing the carbonate-based electrolyte, achieving 87.7% capacity retention (w.r.t. the third cycle) after 200 cycles at 45 °C (vs. 75.7%) and 82.1% retention at 60 °C (vs. 71.4%, Supplementary Fig. 20). In addition to improved high-temperature cycling stability, the EMTMSA-based electrolyte also delivers advantages under thermal-abuse conditions. Pouch cells charged to 100% state of charge (4.2 V) were evaluated by accelerated rate calorimetry (ARC) to track key thermal-runaway temperatures (Fig. 4f). The EMTMSA-based electrolyte increases both the self-generated heat initiation temperature ($T_1$, from 163 to 206 °C) and the thermal-runaway trigger temperature ($T_2$, from 233 to 278 °C), while showing a slight reduction in the maximum thermal-runaway temperature ($T_3$, from 471 to 468 °C). These ARC data verify that the flame-retardant properties of the EMTMSA molecule are effectively propagated to the pouch-cell level, highlighting the value of molecular-level flame-retardant electrolyte design.

## Characterizations of SEIs formed on HC negative electrodes

The pronounced difference in low-temperature cycling performance between the two electrolytes could be primarily attributed to their distinct interactions with the HC negative electrode, which guides us to conduct postmortem analysis on the cycled HC negative electrodes. After 200 cycles at −20 °C, the HC negative electrodes were extracted from the pouch cells in an argon-filled glove box. A large amount of white precipitates was observed on the HC electrode cycled in the carbonate-based electrolyte, indicative of severe $Na^0$ precipitation (Fig. 5a). In contrast, the HC negative electrode cycled in the EMTMSA-based electrolyte retained a uniform black appearance, suggesting a more stable $Na^0$-free interphase (Fig. 5a). The presence of $Na^0$ precipitation was further confirmed by scanning electron microscopy (SEM) coupled with energy-dispersive spectroscopy (EDS) analysis. Mossy-like $Na^0$ dendrites[43–45]–characteristic of sodium metal deposition from carbonate-based electrolytes–were clearly observed on the HC electrode cycled in the carbonate-based electrolyte (Fig. 5b). In contrast, the HC electrode maintained a smooth surface without Na agglomeration after cycling in the EMTMSA-based electrolyte (Supplementary Fig. 21). The suppressed $Na^0$ precipitation observed in the EMTMSA-based electrolyte can be attributed to its lower cell impedance at low temperatures (Supplementary Fig. 22), as well as the formation of a more ionic conductive and stable SEI on the HC negative electrode. While the SEI formed in the carbonate-based electrolyte is approximately 8–10 nm (Fig. 5c), observed in the high-resolution transmission electron microscopy (HRTEM), it is only 2–4 nm (Fig. 5d) in the EMTMSA-based electrolyte. These observations indicate that the two electrolytes should undergo distinct decomposition pathways, despite both having PC as the primary solvent coordinating with $Na^+$. In addition, the polarity differences between the two electrolytes also lead to different SEI solubility. The extent of SEI dissolution was further

evaluated using Na‖Cu cells by monitoring capacity changes after controlled resting intervals according to a previous report[46–48]. An increase in capacity after each pause reflects continued electrolyte decomposition (Fig. 5e), primarily due to the dissolution and reformation of SEI during open-circuit rest. The carbonate-based electrolyte exhibited a larger capacity increase after each pause compared to the EMTMSA-based electrolyte (Fig. 5f), indicating more severe SEI dissolution and inferior ability to suppress parasitic reactions. The higher polarity of the carbonate-based electrolyte may promote continuous dissolution and re-deposition of SEI components, leading to uncontrolled SEI growth and increased interfacial resistance. In contrast, the lower polarity of the EMTMSA-based electrolyte helps preserve SEI integrity and suppress excessive thickening[49].

SEI compositions play an important role in ion transport and dissolution behavior, which was further investigated by surface chemistry analysis. X-ray photoelectron spectroscopy (XPS) results show that the SEI formed on HC in the ETMTMSA-based electrolyte contains a higher proportion of NaF species and a lower proportion of organic species, even including $Na_xC$, which represents dead sodium precipitation, compared to that formed in the carbonate-based electrolyte (Fig. 5g, Supplementary Fig. 23). As a representative inorganic component, NaF contributes to enhancing the chemical robustness of the SEI. This inorganic-rich SEI is less prone to dissolution in organic electrolytes, in contrast to organic-rich SEI components, which tend to dissolve more readily due to the like-dissolves-like principle. To get more accurate insights into the depth distribution of SEI components, Na‖Cu cells were repeatedly cycled within a voltage window of 0.005 to 1.5 V in different electrolytes. This procedure facilitates the formation of a relatively flat and uniform SEI layer on the Cu substrate, which is more suitable for depth profiling compared to the inherently rough and irregular SEI surface typically formed on HC particles. Time-of-flight secondary ion mass spectrometry (TOF−SIMS) was conducted to obtain the depth profiling of different fragments in the SEI. TOF−SIMS was employed to investigate the depth profiles of characteristic SEI fragments. As shown in Fig. 5h, i and Supplementary Fig. 24, the SEIs formed in both electrolytes consist of inorganic ($NaF^-$) and organic ($CHO_2^-$) components, but their content and spatial distributions differ significantly. In the carbonate-based electrolyte, the SEI exhibits a strong $CHO_2^-$ signal and a gradual increase in the $Cu^-$ signal throughout the sputtering process (1000 s), accompanied by a concurrent decrease in $NaF^-$ and $CHO_2^-$ intensities (Fig. 5h). This indicates the formation of a relatively thick SEI. In contrast, the $Cu^-$ signal in the ETMTMSA-derived SEI rapidly saturates within 300 s of sputtering, suggesting a much thinner interphase Fig. 5i. These distinctions are more intuitively visualized in the corresponding 3D reconstructed TOF−SIMS maps (Fig. 5h, i), where the more compact and uniform distribution of $NaF^-$ and $CHO_2^-$ in the ETMTMSA system further corroborates the formation of a thinner and more homogeneous SEI layer, which is consistent with the HRTEM and XPS results.

## Characterizations of CEIs formed on NFM positive electrodes

Operating at high cut-off voltages, such as 4.2 V vs. $Na/Na^+$ (equivalent to 4.5 V vs. $Li/Li^+$), poses a significant challenge to the anodic stability of electrolytes. At these elevated potentials, the TM oxide positive electrode is subjected to both high chemical and mechanical stress. If positive electrode−electrolyte interfacial side reactions are not effectively suppressed, continuous chemical corrosion can occur. Deep charging the layered oxide positive electrode beyond 4.0 $V_{Na}$ often induces excessive Na extraction from the lattice, triggering an O3 to OP2 phase transformation[13] accompanied by the formation of highly oxidative, high-valence Ni species at the positive electrode surface. Under such conditions, intensified parasitic reactions at the CEI promote TM dissolution and oxygen loss, destabilizing the layered framework. The resulting structural degradation, marked by irreversible layer sliding and cation mixing, drives the formation of an

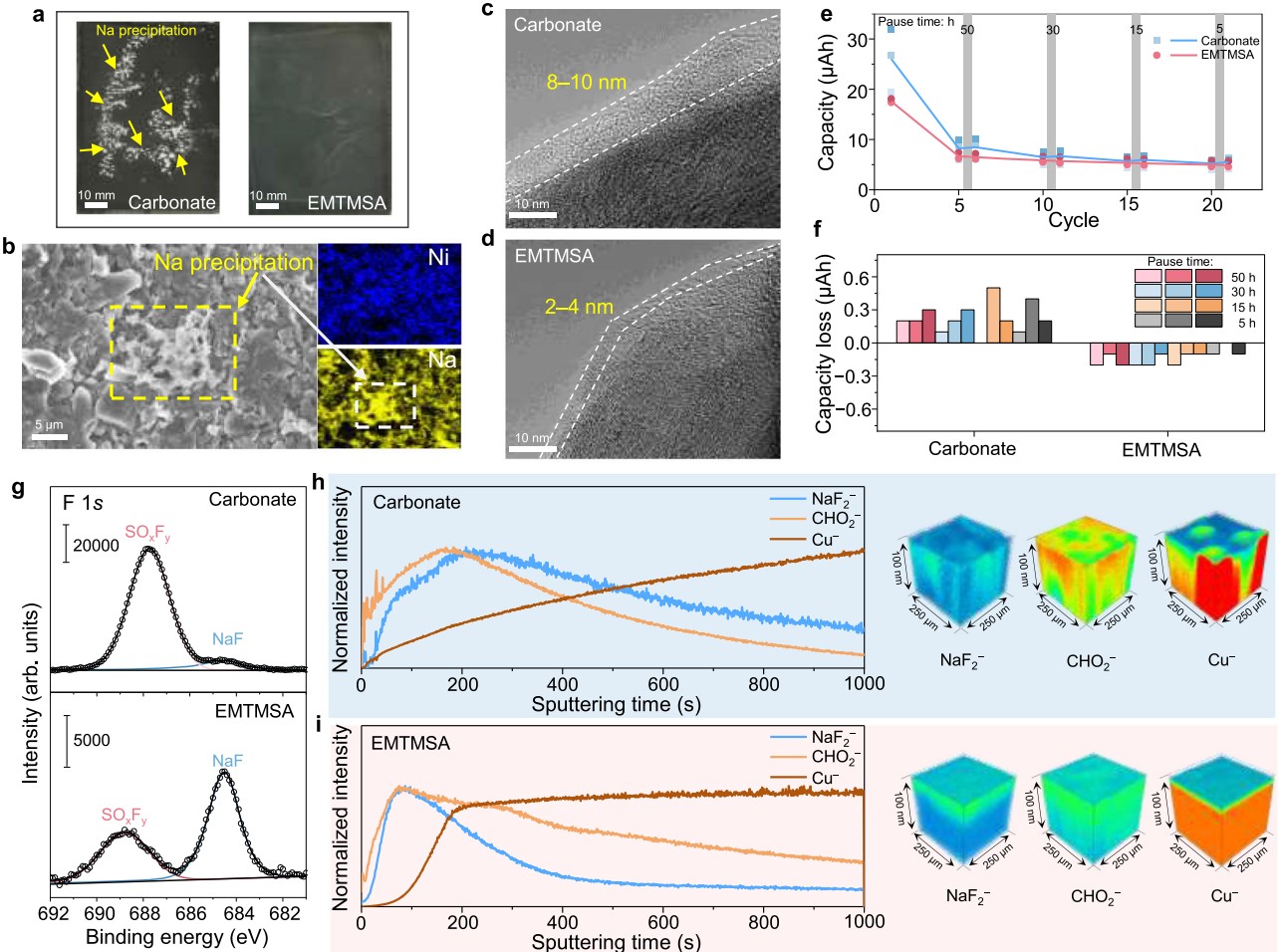

**Fig. 5 | Characterizations of SEIs formed on HC negative electrodes with different electrolytes.** Optical (**a**) and SEM and EDS elemental mapping (**b**) images of the HC negative electrodes extracted from the pouch cells in different electrolytes at −20 °C for 200 cycles. Obvious Na$^0$ precipitation was observed from the one cycled in the carbonate-based electrolyte (**b**). High-resolution TEM images of the HC particles cycled in the carbonate-based (**c**) and ETMTMSA-based (**d**) electrolytes. The dashed line marks the SEI layer (8–10 nm in **c**, and 2–4 nm in **d**). **e** Evolution of the capacity contributed by electrolyte reduction before and after each pause as a function of cycle number by Na||Cu cells with different electrolytes. The gray-shaded regions with arbitrary width indicate the periods during which the cells were paused. The solid lines track the average values from three parallel cells.

**f** Capacity loss associated with different pause durations for cells using carbonate-based and EMTMSA-based electrolytes. Each shaded curve represents an individual cell. **g** F 1$s$ XPS spectra of the SEIs formed in both electrolytes on HC negative electrodes. The intensity scales on the $y$-axis are indicated. TOF−SIMS depth profiles and 3D reconstructed images of the interested species NaF$_2^-$, CHO$_2^-$, and Cu$^-$ signals for SEIs formed in the carbonate-based (**h**) and EMTMSA-based (**i**) electrolytes. Each interested species was normalized based on its maximum signal intensity throughout the entire sputtering time. Blue and red colors correspond to the minimum and maximum intensities, respectively. Source data are provided as a Source data file.

electrochemically inactive, cation-disordered rock-salt phase[50,51], which not only hampers Na$^+$ mobility but also introduces a pronounced stress mismatch with the parent layered framework. Although single-crystalline NFM particles can tolerate higher mechanical stress[34,35] than their polycrystalline counterparts due to the absence of fragile grain boundaries, their effective surface area for Na$^+$ transport is inherently limited. In contrast, polycrystalline particles, despite being mechanically weaker, can expose additional intergranular surfaces upon transgranular cracking, thus increasing Na$^+$ transport pathways. From this perspective, the impedance imposed by the resistive rock-salt layer is more detrimental to single-crystalline NFM, where the restricted surface area magnifies the blocking effect of this passivating phase. After long-term deep cycling, the positive electrode in the carbonate-based electrolyte develops a thick rock-salt reconstruction layer that reaches several tens of micrometers beneath the CEI (Fig. 6a). In contrast, the positive electrode cycled in our EMTMSA-based electrolyte exhibits only a few-micrometer-thick rock-salt region that is discontinuously distributed in isolated nanoscale patches, as outlined by the finely dashed white contours (Fig. 6b). Fast

Fourier transform (FFT) and inverse FFT patterns taken from the marked regions confirm the coexistence of layered and rock-salt structures, allowing direct visualization of the reconstructed rock-salt phase at the surface. Moreover, the CEI formed in the carbonate-based electrolyte is relatively thick (5–10 nm, Fig. 6a), whereas the interphase generated in the EMTMSA-based electrolyte is much thinner at only 2–3 nm (Fig. 6b). This thinner and more compact CEI formed in our EMTMSA-based electrolyte effectively protects the positive electrode surface from further degradation while facilitating Na$^+$ transport, which is consistent with the improved cycling stability observed over 1000 cycles. TOF−SIMS depth profiling of the NFM particles after cycling in the carbonate-based electrolyte reveals that organic components, represented by the CHO$_2^-$ fragment, are present throughout the entire sputtering process (Fig. 6c). Moreover, the CHO$_2^-$ signal intensity is higher than that observed in the CEI formed in the EMTMSA-based electrolyte, as evidenced by both the 2D and 3D reconstructed distribution maps (Fig. 6c, d). In contrast, the decomposition products of the salt NaFSI, such as NaF$^-$ and SO$^-$ fragments, are primarily confined to the surface region for the carbonate-derived

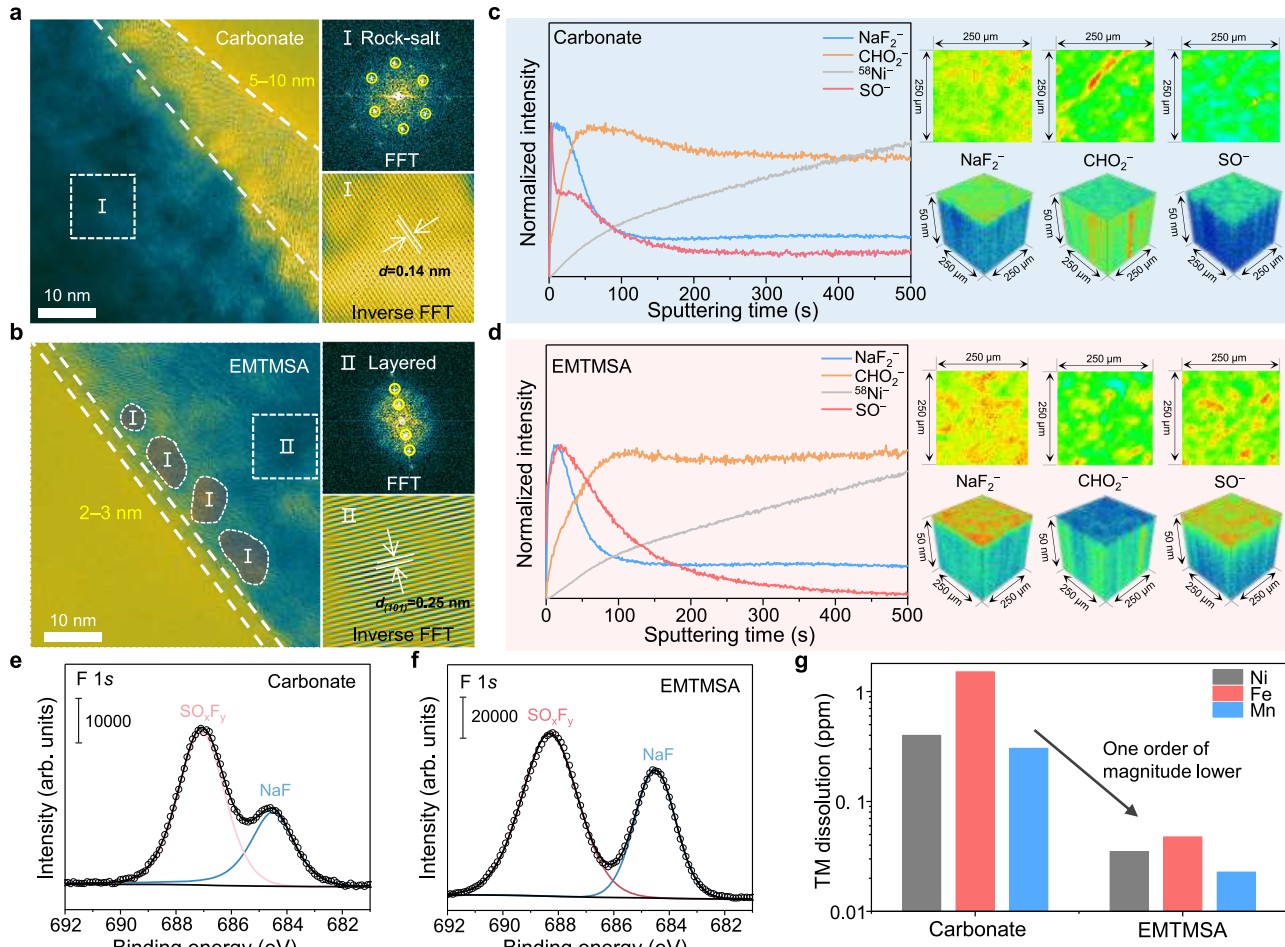

**Fig. 6 | Characterizations of the bulk/surface phase transition and CEIs formed on NFM positive electrodes with different electrolytes.** High-resolution TEM images and corresponding FFT/inverse FFT patterns of the selected areas of the NFM particles cycled in the carbonate-based (**a**) and EMTMSA-based (**b**) electrolytes. Areas I and II indicate rock-salt and layered structures, respectively. The straight dashed line marks the CEI layer (5–10 nm in **a**, and 2–3 nm in **b**). In (**b**), the white, finely dashed contours highlight the thin nanoscale rock-salt regions (I), whereas in (**a**), the entire area beneath the CEI is composed of rock-salt. TOF−SIMS depth profiles and 2D/3D reconstructed images of the interested species for CEIs formed in the carbonate-based (**c**) and EMTMSA-based (**d**) electrolytes. Each interested species was normalized based on its maximum signal intensity throughout the entire sputtering time. Blue and red colors correspond to the minimum and maximum intensities, respectively. F 1s XPS spectra of CEIs formed in the carbonate-based (**e**) and EMTMSA-based (**f**) electrolytes. The intensity scales on the y-axis are indicated. **g** TM dissolution measured by ICP-OES after 1000 cycles in different electrolytes. Source data are provided as a Source data file.

CEI (Fig. 6c). Notably, the intensity of the $SO^-$ fragment decreases rapidly during sputtering, suggesting very limited depth penetration. These results indicate that the majority of the interphase formed in the carbonate-based electrolyte originates from the solvent decomposition rather than the salt. However, for the CEI formed in the EMTMSA-based electrolyte, the $NaF^-$ and $SO^-$ fragments have stronger intensities and a broader spatial distribution (Fig. 6d) compared to those in the carbonate-based electrolyte. This suggests a more extensive incorporation of salt- or EMTMSA-derived species into the CEI, contributing to its inorganic-rich composition. The NaF-rich CEI formed in the EMTMSA-based electrolyte can be further verified by the XPS results (Fig. 6e, f). This reflects the advantage of our EMTMSA molecule with a sulfonimide-salt-like molecular structure that can deduce similar decomposition products as the sulfonimide salt[52] to effectively passivate the high-voltage aggressive positive electrode surface (Supplementary Fig. 25 and Supplementary Data 2). To further characterize the parasitic reaction between the positive electrode and electrolyte, the dissolution of TMs (Ni, Fe, and Mn) after long-term cycling was quantitatively measured by inductively coupled plasma optical emission spectrometry (ICP-OES). Compared with the carbonate-based electrolyte, the EMTMSA-based electrolyte led to an order-of-magnitude reduction in the dissolution of Ni, Fe, and Mn (Fig. 6g).

This further suggests that the CEI formed in the EMTMSA-based electrolyte can effectively mitigate the parasitic reaction and thus suppress the TM dissolution from the NFM positive electrodes.

## Discussion

By introducing asymmetrical alkyl substituents (methyl and ethyl groups) on the sulfonamide nitrogen, we design a non-flammable sulfonamide molecule, EMTMSA, which exhibits a low melting point of −86 °C. It successfully inherits the anodic stability of its structural analog and favors the high-voltage cycling stability. Moreover, the formulated EMTMSA-based electrolyte shows sufficient molecular rotational mobility at low temperatures, as revealed by the long spin-spin relaxation time in VT-NMR. The electrolyte enriched with CIPs and aggregates promotes the formation of stable and low impedance interphases with both the positive electrode and negative electrode, and successfully enables an Ah-level HC‖$NaNi_{1/3}Fe_{1/3}Mn_{1/3}O_2$ pouch cell to maintain 69.8% and 42.3% of its room-temperature capacity even at −60 °C and −70 °C. The pouch cells also exhibit long-term cycling stability with high capacity retentions of 90.0% (cut-off voltage ~4.15 $V_{Na}$) and 81.6% (cut-off voltage ~4.2 $V_{Na}$) after 1500 and 1000 cycles, respectively. The ARC measurements further confirm that the flame-retardant nature of EMTMSA is effectively translated to the

pouch-cell level, elevating the onset and trigger temperatures of thermal runaway and thereby enhancing the intrinsic safety margin of high-voltage SIBs. This work provides a molecular design for practical SIBs under high cut-off voltages and wide temperature.

## Methods

### Materials and electrolyte preparation

Sodium metal (Na) circular disks (12 mm in diameter, 0.45 mm in thickness, 99.7% purity) were obtained from Guangdong Canrd Technology Co. Ltd. Copper (Cu) foil (9 μm in thickness) and aluminum (Al) foil (20 μm in thickness) were purchased from Guangdong Canrd Technology Co. Ltd. and employed after light wiping with anhydrous ethanol, without further processing. NaFSI (99.90%) salt was purchased from Solvionic Co. Ltd. Organic solvents including PC (battery-grade) and EMC (battery-grade) were provided by Guangdong Canrd Technology Co. Ltd. The detailed synthesis method of DMTMSA[53] is as follows: First, 28 mL of 2 M dimethylamine solution in tetrahydrofuran (56.0 mmol), 7.8 mL of triethylamine (56.0 mmol), and 50 mL of dichloromethane (DCM) were added to a 250 mL round-bottomed flask equipped with a magnetic stirring bar. Under a nitrogen atmosphere and at −78 °C, with continuous stirring, trifluoromethanesulfonyl chloride (8.0 g, 47.0 mmol) dissolved in 10 mL DCM was added dropwise to the reaction mixture using a syringe pump. The resulting mixture was stirred at RT overnight. Next, 100 mL of 1 M hydrochloric acid solution was added to separate the organic phase, which was then washed with brine solution (2 × 40 mL). The organic layer was dried over anhydrous sodium sulfate, filtered, and concentrated by rotary evaporation. Finally, the pure colorless liquid product DMTMSA was obtained by distillation twice (115–120 °C, 1 atm). For the synthesis of EMTMSA and DETMSA, dimethylamine was replaced with methylethylamine and diethylamine[54], respectively, as the starting amine, and tetrahydrofuran was not required. All solvents were dried over 4 Å molecular-sieves for 3 days prior to use. The carbonate electrolyte 1 M NaFSI in PC/EMC (3/7, volume ratio, v/v) is used as a reference electrolyte. The sulfonamide-based electrolyte is 1 M NaFSI in EMTMSA/PC/EMC (6/1/3, v/v). Other sulfonamide-based electrolytes were prepared by replacing EMTMSA with DMTMSA or DETMSA, while keeping the composition of 1 M NaFSI in sulfonamide/PC/EMC (6/1/3, v/v) unchanged. Electrolytes were precisely prepared in glass sample vials using a pipette fitted with a polypropylene tip and homogenized by shaking. After use, the vials were sealed with parafilm and stored. All preparation and storage procedures were performed in an argon-filled glove box (both $H_2O$ and $O_2$ levels <0.1 ppm, Shanghai Mikrouna) at 25 °C.

### Pouch cell preparation

Dry Ah-level HC‖NFM pouch cells were purchased from Beijing Bosonlight Technology Co. Ltd. To fabricate the pouch cells, a slurry was first prepared by mixing 95.3 wt% NFM (Jiangsu Xiangying Amperex Technology Co. Ltd.) powder, 2.4 wt% polyvinylidene difluoride (Guangdong Canrd Technology Co. Ltd.), 1.3 wt% Super P (Guangdong Canrd Technology Co. Ltd.) conductive agent, and 1 wt% carbon nanotubes (Guangdong Canrd Technology Co. Ltd.) with N-methylpyrrolidone (battery-grade, Guangdong Canrd Technology Co. Ltd.). For the negative electrode, a slurry was prepared by mixing 94.5 wt% HC powder (Kuarary Co. Ltd.), 2.5 wt% carboxymethyl cellulose (Guangdong Canrd Technology Co. Ltd.), 1 wt% styrene-butadiene rubber (Guangdong Canrd Technology Co. Ltd.), and 2 wt% Super P conductive agent with deionized water. The slurry was homogenized using a mixer machine (AR100, THINKY) at 25 °C in an argon atmosphere. The slurry for the positive electrode and the negative electrode was coated by a doctor-blade method on Al foils, which were dried under vacuum at 120 and 60 °C, respectively. Subsequently, the prepared NFM positive electrodes and HC negative electrodes were punched into rectangular sheets with dimensions of 60 × 80 mm² and

63 × 84 mm², respectively, using a semi-automatic electrode punching machine (MSK-180-S, Shenzhen Kejing Technology Co.). The positive electrode and negative electrode were then carefully stacked layer by layer with Celgard 2320 separator (polypropylene/polyethylene/polypropylene trilayer structure, 20 μm thick, 39% porosity), followed by Al tab welding in an argon-filled glove box at 25 °C. Finally, the dry pouch cells were injected with the electrolyte (6 g Ah⁻¹) and vacuum-sealed. Cell parameters are summarized in Supplementary Table 2. Following electrolyte filling and hot sealing in the glove box, the pouch cells were pre-cycled at 40 °C at a low rate of 0.05 C for the formation step. After formation, the cells underwent a vacuum degassing followed by final sealing (Supplementary Fig. 10). In this process, the edge of the gas pocket is gently punctured to release the accumulated gas under vacuum, and the pouch is then hot-sealed under vacuum to ensure complete removal of residual gas. This process effectively eliminates the influence of formation-induced gas generation on the subsequent cycling tests. A small gas pocket remains after final sealing (Supplementary Fig. 10), the pouch cells are placed in a mechanical fixture (5–6 kg cm⁻²) for cycling tests.

### Electrochemical performance

All cycling tests were carried out using a Neware battery testing system (CT-4008Tn-5V6A-S1), and all the pouch cells were tested inside a battery forced convection thermostat chamber (BTC-706, Shanghai Boling instrument) to ensure a constant temperature under different temperatures, with temperature fluctuations kept within ±0.5 °C of the set value. The cells were initially charged and discharged at 0.1 C for two formation cycles, followed by long-term cycling tests at higher rates. The 1 C rate corresponds to 1 A, equivalent to a current density of approximately 2.60 mA cm⁻² based on the total geometric area of the positive electrode (8 layers × 60 mm × 80 mm = 384 cm²). Moreover, the capacity of the third cycle was used as the initial capacity to calculate the capacity retention. All electrochemical performance tests were conducted without employing constant voltage techniques. Rate capability of the pouch cells with different electrolytes was evaluated at rates of 0.1 C, 0.2 C, 0.5 C, and 1 C for five cycles for each rate between 1.5 V and 4.2 V$_{Na}$. To evaluate the discharge capacities at different low temperatures, the pouch cells with different electrolytes were first charged to 4.2 V$_{Na}$ at 0.1 C at RT, followed by discharging at −20 °C, −40 °C, and −60 °C at 0.1 C. At −70 °C, the cell was discharged at 0.01 C to test its intrinsic discharging performance at such a low temperature. GITT measurements were performed after cycling by repeatedly applying a rate of 0.5 C for 10 min followed by a 50-min rest period. During the constant current pulses, data points were recorded every 10 s, while during the rest periods, data points were recorded every 30 s. EIS during the cell operation, as well as before and after cycling, was performed on a Solartron Energy Lab electrochemical workstation with an amplitude of 5 mV in the frequency range from $10^{-2}$ to $10^5$ Hz. Prior to measurement, the cell was allowed to stabilize at open-circuit voltage for 12 h. The EIS data fitting was performed using a Solartron Energy Lab electrochemical workstation. In situ EIS measurements were conducted using standard pouch cells. The test program was pre-set, and EIS spectra were automatically acquired every 0.1 V during the charge/discharge process. Electrochemical stability window of electrolyte was determined via linear scan voltammetry using Na‖Al cells, scanned from open-circuit voltages to 6 V at a scan rate of 10 mV s⁻¹. Ionic conductivity ($\sigma$) of the electrolyte was measured using steel‖steel symmetric cells at various temperatures. Prior to each measurement, the cells were equilibrated at the designated temperature for at least 1 h to reach temperature balance. Ionic conductivity was then calculated using Eq. (2).

$$\sigma = \frac{L}{SR} \quad (2)$$

Here, $L$ is the distance between the two steel electrodes, $S$ is the surface area of the steel, and $R$ is the measured resistance of the cell. For the SEI dissolution test, Na∥Cu cells were assembled with 50 μL electrolyte. The cells were cycled within a voltage range of 0.005–1.5 V at a constant current density of 2.5 μA cm$^{-2}$ (based on the area of Na electrode) to form the SEI layer on the Cu surface. After every five cycles, the cells were paused at 1.5 V for a period of time to allow SEI dissolution before resuming cycling. Dissolution tests were repeated three times for each electrolyte using parallel Na∥Cu cells to ensure reproducibility. All electrochemical test data were obtained from three parallel cells to ensure reproducibility.

## Characterizations

Melting points of the solvents were determined by DSC using a NETZSCH DSC 3500 instrument with a cooling rate of 10 K min$^{-1}$. The flash point was measured using a closed-cup flash point tester (SYD-3536-1). The dielectric constant was determined using a Wayne Kerr 6530B impedance analyzer. Viscosity was evaluated using an AMETEK Brookfield viscometer at 25 °C. The boiling point of EMTMSA was obtained experimentally during synthesis. The water content of the electrolyte was measured by Karl Fisher titration (Mettler–Toledo). The HF content of the electrolyte was measured by NMR (400 MHz, JEOL). Prior to use, NMR tubes were dried under vacuum at 80 °C. To evaluate the wettability of the electrolytes on the Celgard 2320 separator, contact angle measurements were conducted using a contact angle meter (SDC-200S, SINDIN). Raman spectroscopy (InVia Qontor, Renishaw) was employed to investigate the Raman spectra with an excitation wavelength of 785 nm. VT-NMR was employed to investigate the Na$^+$ solvation structure and interactions in the electrolyte. The targeted electrolytes were carefully transferred into NMR tubes in an argon-filled glove box. Chloroform-$d$ (CDCl$_3$) or D$_2$O was hot-sealed in a capillary tube as an internal reference, which was then coaxially immersed into the NMR tube for locking field in the NMR tests. A cooling system comprising liquid nitrogen and a temperature controller was used to regulate the sample temperature during VT-NMR experiments. Cycled pouch cells were disassembled in an argon-filled glove box with O$_2$/H$_2$O contents <0.1 ppm. The extracted positive electrodes and negative electrodes were then cut into small pieces and rinsed using dimethyl carbonate and diethylene glycol dimethyl ether to remove residual salts, respectively. Following the rinsing process, positive electrodes, negative electrodes, and separators were transferred to the glove box's antechamber and subjected to a vacuum for 12 h to ensure thorough drying. The dried samples were subsequently sealed and stored in an argon-filled glove box for subsequent characterizations. Dissolution of TMs was quantified by ICP-OES (Agilent 5110). Surface chemistry of the cycled electrodes was analyzed by XPS (Thermo Fisher Escalab Xi$^+$) and TOF-SIMS (ION-TOF M6, sputtering beam 250 × 250 μm$^2$, sputtering rate 0.1 nm s$^{-1}$). XPS samples were prepared in a glove box and transferred using a vacuum transfer chamber. TOF-SIMS samples were sealed with aluminum-plastic film inside the glove box and transferred to a dedicated glove box attached to the TOF-SIMS instrument, without any exposure to air throughout the entire process. The morphology of the electrodes was examined using SEM (SU8230, 10 kV) equipped with EDS. HRTEM (JEM-F200) was employed to observe the morphology of the CEI and SEI. SEM and HRTEM samples were also prepared in an argon-filled glove box, sealed with aluminum-plastic film, and stored/transported under an argon atmosphere. They were exposed to air for only a short time (less than 5 s) immediately before insertion into the instrument and evacuation. Thermal runaway study of HC∥NFM pouch cells is conducted using an ARC (BTC130, HEL, England). The pouch cells are charged to 4.2 V at 0.1 C before test. During heat-wait-search mode, the heating step is set as 5 °C per step, and the detection limit is 0.03 °C min$^{-1}$. The thermal runaway point is set as 1 °C min$^{-1}$.

## Computation methods

To explore the annealing and diffusion behaviors of DMTMSA, EMTMSA, and DETMSA molecules, MD simulations were conducted using the GROMACS software suite[55]. The simulations employed the General AMBER Force Field[56] along with AM1-BCC atomic charges. All topological files for the molecules and ions were automatically generated using the AuToFF web server[57]. Simulation boxes containing 400 molecules were initially generated using the Packmol program[58]. After energy minimization, each system underwent a rapid annealing process in which the temperature was decreased from 400 to 240 K over a period of 2 ns, with a time step of 1 ps to facilitate equilibration. Subsequently, the temperature was further reduced from 240 to 160 K over a 160 ns simulation using the velocity-rescale thermostat[59] with a relaxation time constant of 1 ps. The pressure was maintained at $1.01325 \times 10^5$ Pa using the C-rescale barostat with an isothermal compressibility constant of $4.5 \times 10^{-5}$ bar$^{-1}$. Periodic boundary conditions were enforced in all directions. Electrostatic interactions along with van der Waals forces were calculated using the particle-mesh Ewald method, with a cut-off distance set to 15 Å.

A linear fit of the mean square displacement (MSD) of the molecular center of mass was used to calculate the diffusion coefficient, following the Eqs. (3) and (4):

$$MSD(\tau) = < (\mathbf{r}(t+\tau) - \mathbf{r}(t))^2 > \tag{3}$$

$$D_t = \frac{MSD(t)}{6\tau} \tag{4}$$

where $\tau$ represents the lag time between two positions. The coefficient before the MSD takes the value of 1/6 for a three-dimensional system. The lowest unoccupied molecular orbital and HOMO energy levels of the FSI$^-$, NaFSI, DMTMSA, EMTMSA, DETMSA, and Na$^+$–EMTMSA were calculated using the DFT in Gaussian 09 W with B3LYP[60–62]/6-311+g (d, p)[63,64] optimization.

## Data availability

All the data generated in this study are provided in this paper and its Supplementary Information file. The optimized structures of MD and DFT are provided in Supplementary Data 1 and Supplementary Data 2, respectively. Source data are provided with this paper.

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

## Acknowledgements

This study was funded by the National Key R&D Program of China (Grant No. 2022YFB3807800, received by W.X.). The authors are grateful to Pei Zhou at the Instrumental Analysis Center of Xi'an Jiaotong University, Xiaohua Cheng, Yuanbin Qin, Pengcheng Zhang, and Peng Zhang at the Center for Advancing Materials Performance from the Nanoscale (CAMP-Nano) in Xi'an Jiaotong University for characterizations.

## Author contributions

X.C. and W.X. conceived the concept and the project. X. Huang synthesized the solvent. X.C. designed the electrolyte and conducted electrochemical measurements. X.C. conducted Raman, SEM, TEM, XPS, and TOF-SIMS measurements and analyzed the results. X.C. and G.C. conducted VT-NMR measurements and analyzed the results. X.C. and L.H. conducted ARC measurements and analyzed the results. X.C. and X. Han conducted MD and DFT calculations and analyzed the results. X.C., Q.L., and W.T. conducted other characterizations. X.C. and W.X. wrote and revised the manuscript. All authors discussed the results and reviewed the manuscript.

## Competing interests

The authors declare no competing interests.
