## [Transparent Peer Review file · Nature Communications]

Asymmetric sulfonamide design enabling high-voltage sodium-ion pouch cells in wide temperature

Corresponding Author: Professor Weijiang Xue

Version 0:

Reviewer comments:

Reviewer #1

(Remarks to the Author)

In the present work, the authors rationally designed a novel sulfonamide molecule bearing asymmetric alkyl groups, which exhibits an impressively low melting temperature down to $-86\text{ }^{\circ}\text{C}$. For such a polar molecule, it is remarkable that the authors successfully balanced polarity, melting temperature, and anodic stability simultaneously. The discussion on the solvation structure and the formulation of the electrolyte is well organized and convincing. From a practical perspective, this work is also timely, as sodium-ion batteries are approaching the stage of industrialization, which requires to further improve their energy density and performance under extreme conditions. Notably, the authors carried out comprehensive electrochemical evaluations using Ah-scale pouch cells, which convincingly demonstrate the superior overall performance of the designed electrolyte. The results highlight excellent cycling stability at high voltages (4.2 V) and superior discharging performance even under extremely low temperatures. Therefore, I recommend to accept it after addressing the following comments.

1. In Fig. 1d, the authors measured the freezing point of the DMTMSA molecule by DSC. It is also recommended to further supplement its practical electrochemical performance of the DMTMSA-based electrolyte prepared with this molecule at low temperatures.
2. The authors demonstrate that the as-prepared EMTMSA-based electrolyte is flame retardant (Fig. 1h). To more comprehensively highlight its flame-retardant characteristics, it is recommended to provide the video of the electrolyte ignition test in as supplementary materials.
3. Pouch cells using the EMTMSA-based electrolyte still maintain an impressive capacity retention even at extremely low temperatures, and exhibit excellent long-term cycling stability at $-20\text{ }^{\circ}\text{C}$, fully demonstrating the outstanding low-temperature performance of this electrolyte. However, from a practical application perspective, how about its high-temperature electrochemical performance?
4. To better meet the requirements of practical applications, it is also recommended to provide the rate performance of cells with this electrolyte.
5. The impedance evolution of pouch cells upon cycling is also important to evaluate electrolyte. Please conduct EIS analysis on the pouch cells before and after cycling from a kinetic perspective.
6. The authors performed TEM analyses on the formed CEIs after cycling in different electrolytes. It is also required to give a more detailed analysis of surface phase transition of the cathode as it is a main degradation mechanism for layered oxide cathodes.

Reviewer #2

(Remarks to the Author)

This work offers some fundamental insights into solvent molecule and electrolyte design for advancing high-energy and low-temperature sodium-ion batteries. It is indicated that the EMTMSA-based electrolyte can enable 1-Ah-level $\text{NaNi}_{1/3}\text{Fe}_{1/3}\text{Mn}_{1/3}\text{O}_2$ ||hard carbon pouch cells to retain 69.8% and 42.3% of room-temperature capacity even at $-60\text{ }^{\circ}\text{C}$ and $-70\text{ }^{\circ}\text{C}$, while achieving outstanding capacity retentions of 90.0% and 81.6% after 1500 and 1000 cycles at high upper cut-off voltages of 4.15 V and 4.2 V, respectively. This paper presents some interesting results. Therefore, it is recommended to be accepted. However, before acceptance, a number of questions need to be carefully addressed and relevant revisions are mandatory:

1. In the experimental section, all solvents were dried over 4 \AA molecular sieves for 3 days prior to use. Although it is pointed

out that the water could be removed, from a comprehensive perspective, it is suggested that the water and impurity contents in the electrolytes should be measured and provided in order to evaluate the performance of the electrolyte.

2. The cycled pouch cells were disassembled in the glovebox, and the cathodes, anodes, and separator were extracted for subsequent characterizations, those information on how to prepare test samples and ensure their intrinsic characteristics should be presented.

3. Both the DMTMSA- and EMTMSA-based electrolytes exhibit similarly high electrochemical stability up to 4.2 V_{Na}. The author explained that the gradual elongation of the alkyl substituents leads to a decrease in the oxidative stability of the electrolyte, likely due to the changed electron-donating effect that compromises anodic resistance. How to verify this conjecture?

4. EMTMSA-based electrolyte demonstrates largely improved flame retardancy, effectively suppressing ignition and sustaining non-flammability. Does this electrolyte have a negative impact on battery performance, such as capacity and coulombic efficiency? If there are negative impacts, how to strike a balance.

5. The safety of batteries is related to the self-generated heat initiation temperature T₁, the thermal runaway trigger temperature T₂, and the maximum thermal runaway temperature T₃. Whether the EMTMSA-based electrolyte can improve battery safety by changing T₁, T₂, and T₃? Please provide relevant evidence.

6. In the battery cycle test depicted in Figure 3a and 3b, the airbag of the pouch cell is not removed. For the control group cell, gas is produced during the formation stage. If the gas is not completely removed, it will also have a negative impact on the battery cycle test, making it impossible to determine the effect of functional electrolyte components.

7. The use of EMTMSA-based electrolyte cell demonstrates excellent low-temperature performance, but the high-temperature performance of the cell is not mentioned in the article. How about the cycling performance at 45 °C and 60 °C?

8. For electrolyte solvents, physicochemical parameters such as dielectric constant, melting and boiling points, flash point, and viscosity are crucial. Please provide the relevant parameters of EMTMSA to demonstrate its potential as a promising electrolyte solvent for sodium-ion batteries.

9. The author investigated the unique effects of several components, including DMTMSA, DETMSA, and EMTMSA, in different battery systems. Can these components also exhibit similar functional characteristics in the same battery material system?

Reviewer #3

(Remarks to the Author)

This manuscript introduces a novel non-flammable, low-melting-point sulfonamide molecule, EMTMSA, as a solvent for sodium-ion battery electrolytes. The system demonstrates fast sodium-ion transport and stable electrode–electrolyte interphase formation. The authors conducted extensive material characterizations and evaluated electrochemical performance in Ah-level pouch cells, which showed promising high-voltage and low-temperature performance with potential for practical applications. However, several conclusions and discussions require clarification or further justification. I therefore recommend a major revision before this manuscript can be considered for publication. Specific comments are as follows:

1. What is the relative percentage of SSIP, CIP, and AGG in both EMTMSA- and carbonate-based electrolytes? The manuscript states that AGG and CIP dominate in the EMTMSA-based electrolyte, yet the solvation structure in Fig. 2f does not show any FSI⁻ anions.
2. For low-temperature operation of SIBs, charging is usually much more difficult than discharging. Since Figs. 4a and 4b only display LT discharge capacity, can the cells also be charged under the same conditions? Long-term viability at extreme temperatures should also be demonstrated.
3. The authors mentioned that the suppressed Na precipitation observed in the EMTMSA-based electrolyte can be attributed to its lower cell impedance at low temperatures. However, the evolution in cell impedance at LT is not provided.
4. How to interpret the “negative” capacity losses associated with EMTMSA electrolyte in Fig. 5f.
5. SEI formed on hard carbon and Cu substrates can differ in chemical composition and thickness. It is recommended to include SEI characterization of hard carbon anodes after cycling.
6. What sputtering rate was used in TOF-SIMS? Some experimental details are missing in the Supporting Information and should be clarified.
7. Given the structural similarity between NaFSI and EMTMSA, how can one distinguish whether the inorganic SEI/CEI components originate from salt or EMTMSA decomposition? Does this imply that EMTMSA also contributes to interfacial stability via sacrificial decomposition? A DFT analysis of HOMO–LUMO levels may help identify decomposition preference.

Reviewer #4

(Remarks to the Author)

Version 1:

Reviewer comments:

Reviewer #1

(Remarks to the Author)

The authors have satisfactorily addressed the reviewer's comments, and the revisions have strengthened the manuscript overall.

Reviewer #2

(Remarks to the Author)

The manuscript has been revised according to reviewers' comments and suggestion.

Reviewer #3

(Remarks to the Author)

The authors have addressed my questions.

Reviewer #4

(Remarks to the Author)

Point-to-point response

Reviewer #1 (Remarks to the Author):

In the present work, the authors rationally designed a novel sulfonamide molecule bearing asymmetric alkyl groups, which exhibits an impressively low melting temperature down to $-86\text{ }^{\circ}\text{C}$. For such a polar molecule, it is remarkable that the authors successfully balanced polarity, melting temperature, and anodic stability simultaneously. The discussion on the solvation structure and the formulation of the electrolyte is well organized and convincing. From a practical perspective, this work is also timely, as sodium-ion batteries are approaching the stage of industrialization, which requires to further improve their energy density and performance under extreme conditions. Notably, the authors carried out comprehensive electrochemical evaluations using Ah-scale pouch cells, which convincingly demonstrate the superior overall performance of the designed electrolyte. The results highlight excellent cycling stability at high voltages (4.2 V) and superior discharging performance even under extremely low temperatures. Therefore, I recommend to accept it after addressing the following comments.

Response: We appreciate Reviewer #1 for the positive comments on our work.

Comment 1: In Fig. 1d, the authors measured the freezing point of the DMTMSA molecule by DSC. It is also recommended to further supplement its practical electrochemical performance of the DMTMSA-based electrolyte prepared with this molecule at low temperatures.

Response: We sincerely thank the reviewer for this valuable suggestion. To validate the practical low-temperature performance of the DMTMSA-based electrolyte, we evaluated the discharge performance of the Ah-level NFM||HC pouch cell with the DMTMSA-based electrolyte at room temperature to $-60\text{ }^{\circ}\text{C}$ (Fig. R1). At $-20\text{ }^{\circ}\text{C}$ and $-40\text{ }^{\circ}\text{C}$, the discharge capacity of the pouch cell utilizing the DMTMSA-based electrolyte reached 90.1% and 79.8% of their room temperature capacity, respectively. This performance is slightly lower compared to the EMTMSA-based electrolyte's pouch cell (which retained 93.8% and 80.6% of capacity at the same temperatures). At a lower temperature of $-60\text{ }^{\circ}\text{C}$, the DMTMSA-based pouch cell delivered only 0.2% of its room-temperature discharge capacity, whereas the EMTMSA-based electrolyte retained 69.8%. One possible reason is that the melting point of the DMTMSA molecule is only $-60\text{ }^{\circ}\text{C}$, much higher than that of EMTMSA ($-86\text{ }^{\circ}\text{C}$), limiting its fluidity and ion transport at ultra-low temperatures. Fig. R1 has been updated in the Supplementary Information, and the related discussion in the main text has been revised accordingly. All modifications are highlighted in yellow.

Fig. R1 Voltage profiles of the pouch cell with the DMTMSA-based electrolyte during discharging at 25 °C, -20 °C, -40 °C, and -60 °C.

Changes:

Main text:

“Similarly, the capacity of the pouch cell using the DMTMSA-based electrolyte declines to nearly zero at -60 °C (Supplementary Fig. 17).”

Supplementary Information:

Updated Supplementary Fig. 17 Voltage profiles of the pouch cell with the DMTMSA-based electrolyte during discharging at 25 °C, -20 °C, -40 °C, and -60 °C.

Comment 2: The authors demonstrate that the as-prepared EMTMSA-based electrolyte is flame retardant (Fig. 1h). To more comprehensively highlight its flame-retardant characteristics, it is recommended to provide the video of the electrolyte ignition test in as supplementary materials.

Response: We appreciate the reviewer’s comments. To further visualize the flame-retardant property of the EMTMSA-based electrolyte, we have uploaded the videos of the ignition tests for both electrolytes as supplementary videos. These videos clearly contrast the rapid

combustion of the carbonate-based electrolyte with the flame retardancy of our EMTMSA-based electrolyte. These videos have been included in the Supplementary information as Supplementary Videos S1 and S2.

Changes:

Main text:

“Furthermore, we performed ignition tests by exposing both electrolytes to an open flame (Fig. 1h, **Supplementary Video S1 and S2**).”

Comment 3: Pouch cells using the EMTMSA-based electrolyte still maintain an impressive capacity retention even at extremely low temperatures, and exhibit excellent long-term cycling stability at $-20\text{ }^{\circ}\text{C}$, fully demonstrating the outstanding low-temperature performance of this electrolyte. However, from a practical application perspective, how about its high-temperature electrochemical performance?

Response: We appreciate the reviewer’s comments, which highlights the importance of the high-temperature stability. We evaluated the electrochemical performance of pouch cells using both the carbonate-based and EMTMSA-based electrolytes at high temperatures ($45\text{ }^{\circ}\text{C}$ and $60\text{ }^{\circ}\text{C}$). Pouch cells with the EMTMSA-based electrolyte exhibit better cycling stability at elevated temperatures compared to those with the carbonate-based electrolyte. After 200 cycles at $45\text{ }^{\circ}\text{C}$, the capacity retention improves from 75.7% (carbonate-based electrolyte) to 87.7% (EMTMSA-based electrolyte), and at $60\text{ }^{\circ}\text{C}$, from 71.4% to 82.1% (Fig. R2). This result further indicates the advantage of our electrolyte for wide-temperature operation and practical applications.

Fig. R2 Cycling performance of pouch cells with the carbonate-based and EMTMSA-based electrolytes at high temperatures with an upper cut-off voltage of 4.2 V_{Na} at 0.5 C . (a) $45\text{ }^{\circ}\text{C}$. (b) $60\text{ }^{\circ}\text{C}$.

Changes:

Main text:

“Moreover, the high-temperature stability at 45 °C and 60 °C was evaluated in pouch cells at an upper charging voltage of 4.2 V_{Na}. Pouch cells with the EMTMSA-based electrolyte exhibit better cycling stability compared with those employing the carbonate-based electrolyte, achieving 87.7% capacity retention after 200 cycles at 45 °C (vs. 75.7%) and 82.1% retention at 60 °C (vs. 71.4%, Supplementary Fig. 20).”

Supplementary Information:

Updated Supplementary Fig. 20 Cycling performance of pouch cells with the carbonate-based and EMTMSA-based electrolytes at high temperatures with an upper cut-off voltage of 4.2 V_{Na} at 0.5 C. (a) 45 °C. (b) 60 °C.

Comment 4: To better meet the requirements of practical applications, it is also recommended to provide the rate performance of cells with this electrolyte.

Response: We appreciate the reviewer’s comment. Rate performance of the NFM||HC pouch cells with the carbonate-based and EMTMSA-based electrolytes was evaluated (Fig. R3, 1 C = 1 A). The cell with the EMTMSA-based electrolyte exhibited 0.990, 0.956, 0.914, and 0.863 Ah at 0.1 C, 0.2 C, 0.5 C, and 1 C, respectively, whereas the carbonate-based cell delivered 0.951, 0.828, 0.713, and 0.632 Ah at the corresponding rates. It is noted that the cell with our EMTMSA-based electrolyte showed better rate capability than the carbonate-based electrolyte.

Fig. R3 Rate performance of the pouch cells using the carbonate-based and EMTMSA-based electrolytes at different rates and upper cut-off voltage of $4.2 V_{Na}$. (a) Comparison of discharge capacity of the pouch cells at different rates. Corresponding voltage profiles for the (b) carbonate-based and (c) EMTMSA-based electrolytes.

Changes:

Main text:

“Compared to the carbonate-based electrolyte, the pouch cell with the EMTMSA-based electrolyte also delivers improved average Coulombic efficiency (CE) at $4.2 V_{Na}$ ($\sim 99.94\%$, Supplementary Fig. 13) and better rate performance (Supplementary Fig. 15).”

Supplementary Information:

Updated Supplementary Fig. 15 Rate performance of the pouch cells using the carbonate-based and EMTMSA-based electrolytes at different rates and upper cut-off voltage of 4.2 V_{Na}. (a) Comparison of discharge capacity of the pouch cells at different rates. Corresponding voltage profiles for the (b) carbonate-based and (c) EMTMSA-based electrolytes.

Experimental section:

“Rate capability of the pouch cells with different electrolytes was evaluated at rates of 0.1 C, 0.2 C, 0.5 C, and 1 C for five cycles for each rate between 1.5 V and 4.2 V_{Na}.”

Comment 5: The impedance evolution of pouch cells upon cycling is also important to evaluate electrolyte. Please conduct EIS analysis on the pouch cells before and after cycling from a kinetic perspective.

Response: We agree with the reviewer’s comment on the evolution of impedance, which can give us more information on the kinetics. As suggested, we performed EIS analysis on the pouch cells using both electrolytes before and after 1000 cycles at an upper cut-off voltage of 4.2 V_{Na}. The Nyquist plots, along with the equivalent circuit model and fitted parameters, are presented in Fig. R4. The impedance from the electrolyte (R_e), corresponding to the high-frequency intercept with the x -axis, dramatically increased from 279.9 m Ω to 334.8 m Ω in the carbonate-based electrolyte, but only marginally grew from 183.1 m Ω to 212.3 m Ω in the EMTMSA-based electrolyte. After long-term cycling, the interphase impedance ($R_{SEI/CEI}$) increased in both electrolytes. However, the carbonate-based electrolyte showed a pronounced rise from 85.4 to 155.6 m Ω (nearly a two-fold increase), whereas the EMTMSA-based electrolyte exhibited only a slight increase from 12.9 to 16.6 m Ω . These results indicate that the EMTMSA-based electrolyte establishes a mechanically and chemically stable interphase on both electrodes, enabling reliable operation at a high cutoff voltage of 4.2 V_{Na}.

Fig. R4 The Nyquist plots, along with the equivalent circuit model and parameters (insets), fitted from EIS analysis on the NFM||HC pouch cells using the (a) carbonate-based and (b) EMTMSA-based electrolytes before and after long-term cycling at 4.2 V_{Na}.

Changes:

Main text:

“These results can also be verified by the impedance analysis on the pouch cells before and after long-term cycling (Supplementary Fig. 16).”

Supplementary Information:

Supplementary Fig. 16 The Nyquist plots, along with the equivalent circuit model and parameters (insets), fitted from EIS analysis on the NFM||HC pouch cells using the (a) carbonate-based and (b) EMTMSA-based electrolytes before and after long-term cycling at 4.2 V_{Na}.

Experimental section:

“Electrochemical impedance spectroscopy (EIS) during the cell operation, as well as before and after cycling, were performed on a Solartron Energy Lab electrochemical workstation with an amplitude of 5 mV in the frequency range from 10⁻² to 10⁵ Hz.”

Comment 6: The authors performed TEM analyses on the formed CEIs after cycling in different electrolytes. It is also required to give a more detailed analysis of surface phase transition of the cathode as it is a main degradation mechanism for layered oxide cathodes.

Response: We agree with the reviewer’s comment and include a more detailed discussion on the surface degradation. Deep charging the layered oxide cathode beyond 4.0 V_{Na} often induces

“excessive” Na extraction from the lattice, triggering a O3 to OP2 phase transformation (*Adv. Mater.* 37, 2415611 (2025)) accompanying by the formation of highly oxidative, high-valence Ni species at the cathode surface. Under such conditions, intensified parasitic reactions at the cathode–electrolyte interface promote transition-metal dissolution and oxygen loss, destabilizing the layered framework. The resulting structural degradation, marked by irreversible layer sliding and cation mixing, drives the formation of an electrochemically inactive, cation-disordered rock-salt phase (*J. Mater. Chem. A* 12, 12443–12451 (2024); *Mater. Today* 89, 35–43 (2025)), which not only hampers Na⁺ mobility but also introduces a pronounced stress mismatch with the parent layered framework. Although single-crystalline NFM particles can tolerate higher mechanical stress than their polycrystalline counterparts due to the absence of fragile grain boundaries, their effective surface area for Na⁺ transport is inherently limited. In contrast, polycrystalline particles, despite being mechanically weaker, can expose additional intergranular surfaces upon transgranular cracking, thus increasing Na⁺ transport pathways. From this perspective, the impedance imposed by the resistive rock-salt layer is more detrimental to single-crystalline NFM, where the restricted surface area magnifies the blocking effect of this passivating phase. After long-term deep cycling, the cathode in the carbonate-based electrolyte develops a thick rock-salt reconstruction layer that reaches several tens of micrometers beneath the CEI (Fig. R5a). In contrast, the cathode cycled in our EMTMSA-based electrolyte exhibits only a few-micrometer-thick rock-salt region that is discontinuously distributed in isolated nanoscale patches, as outlined by the finely dashed white contours (Fig. R5b). Fast Fourier transform (FFT) and inverse FFT patterns taken from the marked regions (Fig. R5) confirm the coexistence of layered and rock-salt structures, allowing direct visualization of the reconstructed rock-salt phase at the surface. Moreover, the CEI formed in the carbonate-based electrolyte is relatively thick (5–10 nm, Fig. R5a), whereas the interphase generated in the EMTMSA-based electrolyte is much thinner at only 2–3 nm (Fig. R5b). This thinner and more compact CEI formed in our EMTMSA-based electrolyte effectively protects the cathode surface from further degradation while facilitating Na⁺ transport, which is consistent with the markedly improved cycling stability observed over 1000 cycles.

Fig. R5 High-resolution TEM images and corresponding FFT/inverse FFT patterns of the selected areas of the NFM particles cycled in the carbonate-based (a) and EMTMSA-based (b) electrolytes. Scale bars, 10 nm. Area I and II indicate rock-salt and layered structures, respectively. The straight dashed line marks the CEI layer (5–10 nm in a, and 2–3 nm in b). In (b), the white finely dashed contours highlight the thin nanoscale rock-salt regions (I), whereas in (a) the entire area beneath the CEI is composed of rock-salt.

Changes:

Main text:

“Deep charging the layered oxide cathode beyond 4.0 V_{Na} often induces “excessive” Na extraction from the lattice, ... This thinner and more compact CEI formed in our EMTMSA-based electrolyte effectively protects the cathode surface from further degradation while facilitating Na⁺ transport, which is consistent with the markedly improved cycling stability observed over 1000 cycles.”

Updated Fig. 6 Characterizations of the bulk/surface phase transition and CEIs formed on NFM cathodes with different electrolytes. High-resolution TEM images and corresponding FFT/inverse FFT patterns of the selected areas of the NFM particles cycled in the carbonate-based (a) and EMTMSA-based (b) electrolytes. Scale bars, 10 nm. Area I and II indicate rock-salt and layered structures, respectively. The straight dashed line marks the CEI layer (5–10 nm in a, and 2–3 nm in b). In (b), the white finely dashed contours highlight the thin nanoscale rock-salt regions (I), whereas in (a) the entire area beneath the CEI is composed of rock-salt. TOF-SIMS depth profiles and 2D/3D reconstructed images of the interested species for CEIs formed in the carbonate-based (c) and EMTMSA-based (d) electrolytes. Scale bars, 25 μ m. Blue and red colors correspond to the minimum and maximum intensities respectively. F1s XPS spectra of CEIs formed in the carbonate-based (e) and EMTMSA-based (f) electrolytes. (g) TM dissolution measured by ICP-OES after 1000 cycles in different electrolytes.

Reference:

“49. Zhao, X. *et al.* Insights into the capacity fading and failure mechanism of an O3-NaNi_{1/3}Fe_{1/3}Mn_{1/3}O₂ layered oxide cathode material for sodium-ion batteries. *J. Mater. Chem. A* **12**, 12443–12451 (2024).”

“50. Liu, M. *et al.* Layered-to-rocksalt atomic reconfiguration on O3-type cathodes surface for high-energy and durable sodium-ion batteries. *Mater. Today* **89**, 35–43 (2025).”

Reviewer #2 (Remarks to the Author):

This work offers some fundamental insights into solvent molecule and electrolyte design for advancing high-energy and low-temperature sodium-ion batteries. It is indicated that the EMTMSA-based electrolyte can enable 1-Ah-level $\text{NaNi}_{1/3}\text{Fe}_{1/3}\text{Mn}_{1/3}\text{O}_2$ ||hard carbon pouch cells to retain 69.8% and 42.3% of room-temperature capacity even at $-60\text{ }^\circ\text{C}$ and $-70\text{ }^\circ\text{C}$, while achieving outstanding capacity retentions of 90.0% and 81.6% after 1500 and 1000 cycles at high upper cut-off voltages of 4.15 V and 4.2 V, respectively. This paper presents some interesting results. Therefore, it is recommended to be accepted. However, before acceptance, a number of questions need to be carefully addressed and relevant revisions are mandatory:

Response: We appreciate the reviewer's recognition and positive comments on our work.

Comment 1: In the experimental section, all solvents were dried over 4 Å molecular sieves for 3 days prior to use. Although it is pointed out that the water could be removed, from a comprehensive perspective, it is suggested that the water and impurity contents in the electrolytes should be measured and provided in order to evaluate the performance of the electrolyte.

Response: We appreciate the reviewer's comment on the detailed water content, which is important for electrolyte quality. The most common impurities in electrolyte are water and hydrogen fluoride (HF). Accordingly, the water content was assessed by Karl Fischer titration (Mettler–Toledo), and the HF content was analyzed by NMR for the EMTMSA-based electrolyte (Fig. R6). The EMTMSA-based electrolyte contained 67 ppm of water prior to drying, which was markedly reduced to 26 ppm after treating the solvent with 4 Å molecular sieves for 3 days (Fig. R6a). ^{19}F NMR (Fig. R6b) and ^1H NMR (Fig. R6c) data of the EMTMSA-based electrolyte after molecular-sieve drying show no detectable HF impurities. The low impurity level in our electrolyte is essential for achieving its high electrochemical performance.

Fig. R6 Characterizations of impurity in the EMTMSA-based electrolyte. (a) Water content of the EMTMSA-based electrolyte prepared with solvents not subjected to 4 Å molecular-sieve dehydration (left bar) and with solvents dehydrated over 4 Å molecular-sieves for 3 days (right bar). (b) ^{19}F NMR and (c) ^1H NMR of the dehydrated EMTMSA-based electrolyte. The molecular structures in (b) and (c), annotated with circles and arrows, indicate the peak assignments in the ^{19}F and ^1H NMR spectra. All observed peaks are fully assigned, and no signals appear in the HF chemical-shift region, confirming the absence of detectable HF.

Changes:

Main text:

“All solvents were subjected to molecular-sieve drying to eliminate residual water and consequently suppress hydrogen fluoride (HF) generation (Supplementary Fig. 2), which is vital for maintaining stable electrochemical performance.”

Supplementary Information:

Updated Supplementary Fig. 2 Characterizations of impurity in the EMTMSA-based electrolyte. (a) Water content of the EMTMSA-based electrolyte prepared with solvents not subjected to 4 Å molecular-sieve dehydration (left bar) and with solvents dehydrated over 4 Å molecular-sieves for 3 days (right bar). (b) ^{19}F NMR and (c) ^1H NMR of the dehydrated EMTMSA-based electrolyte. The molecular structures in (b) and (c), annotated with circles and arrows, indicate the peak assignments in the ^{19}F and ^1H NMR spectra. All observed peaks are fully assigned, and no signals appear in the HF chemical-shift region, confirming the absence of detectable HF.

Experimental section:

“The water content of the electrolyte was measured by Karl Fisher titration (Mettler–Toledo). The hydrogen fluoride (HF) content of the electrolyte was measured by nuclear magnetic resonance (NMR, 400 MHz, JEOL). Prior to use, NMR tubes were dried under vacuum at 80 °C.”

Comment 2: The cycled pouch cells were disassembled in the glovebox, and the cathodes, anodes, and separator were extracted for subsequent characterizations, those information on how to prepare test samples and ensure their intrinsic characteristics should be presented.

Response: We appreciate your comments on the preparation of the sensitive samples in glove box. Cycled pouch cells were disassembled in an argon-filled glovebox with $\text{O}_2/\text{H}_2\text{O}$ contents

<0.1 ppm. The extracted cathodes and anodes were then cut into small pieces and rinsed using dimethyl carbonate (DMC) and diethylene glycol dimethyl ether (DEGDME) to remove residual salts, respectively. Following the rinsing process, cathodes, anodes, and separators were transferred to the glovebox's antechamber and subjected to vacuum for 12 h to ensure thorough drying. The dried samples were subsequently sealed and stored in the glovebox for subsequent characterizations. We have supplemented the Experimental section of the manuscript with detailed steps for sample preparation:

Changes:

Experimental section:

“Cycled pouch cells were disassembled in an argon-filled glovebox with O₂/H₂O contents <0.1 ppm. The extracted cathodes and anodes were then cut into small pieces and rinsed using dimethyl carbonate (DMC) and diethylene glycol dimethyl ether (DEGDME) to remove residual salts, respectively. Following the rinsing process, cathodes, anodes, and separators were transferred to the glovebox's antechamber and subjected to vacuum for 12 h to ensure thorough drying. The dried samples were subsequently sealed and stored in the glovebox for subsequent characterizations.”

Comment 3: Both the DMTMSA- and EMTMSA-based electrolytes exhibit similarly high electrochemical stability up to 4.2 V_{Na}. The author explained that the gradual elongation of the alkyl substituents leads to a decrease in the oxidative stability of the electrolyte, likely due to the changed electron-donating effect that compromises anodic resistance. How to verify this conjecture?

Response: We appreciate the reviewer's comment on the molecular design. As the alkyl chain length increases, its greater electron-donating capacity attenuates the electron-withdrawing effect of the FSO₂⁻ group on the terminal substituent, resulting in a higher local electron density compared with a -CH₃ terminus. This mechanistic trend is supported consistently by both electrochemical measurements and theoretical calculations.

On the one hand, linear sweep voltammetry (LSV) was performed using Na||Al cells to evaluate the anodic decomposition onset potentials of the three electrolytes. As shown in Fig. R7a (previous Supplementary Fig. 2), the DETMSA-based electrolyte exhibits the lowest onset potential (~4.1 V), whereas the DMTMSA- and EMTMSA-based electrolytes show higher values (~4.8 V). This trend experimentally supports the analysis that elongating the alkyl substituent diminishes oxidative stability.

On the other hand, density functional theory (DFT) calculations were carried out to compare the frontier molecular orbitals of DMTMSA, EMTMSA, and DETMSA. The highest occupied molecular orbital (HOMO) energy levels show a clear monotonic increase with the gradual extension of the alkyl group (-8.0319 eV for DMTMSA, -7.8983 eV for EMTMSA, and -7.7963 eV for DETMSA; Fig. R7b), directly indicating that the enhanced electron-donating effect renders the molecule thermodynamically more prone to oxidation.

Fig. R7 The relationship between oxidative stability and the length of the alkyl substituent of the three sulfonamide derivatives DMTMSA, EMTMSA, and DETMSA. (a) Electrochemical stability window of different electrolytes via LSV. (b) LUMO and HOMO energy levels of the DMTMSA, EMTMSA, and DETMSA calculated by DFT.

Changes:

Main text:

“The DMTMSA- and EMTMSA-based electrolytes maintain high oxidative stability up to ≈ 4.8 V_{Na}, whereas the DETMSA-based electrolyte exhibits a greatly reduced anodic stability (Supplementary Fig. 3a). This decrease correlates with the substituent’s electronic structure: longer alkyl chains donate more electron density, weakening the FSO₂⁻ group’s electron-withdrawing effect and elevating the local electron density, consistent with the density functional theory (DFT) results showing higher highest occupied molecular orbital (HOMO) levels for longer chains (Supplementary Fig. 3b).”

Supplementary Information:

Updated Supplementary Fig. 3 The relationship between oxidative stability and the length of the alkyl substituent of the three sulfonamide derivatives DMTMSA, EMTMSA, and DETMSA. (a) Electrochemical stability window of different electrolytes via LSV. (b) LUMO and HOMO energy levels of the DMTMSA, EMTMSA, and DETMSA calculated by DFT.

Experimental section:

“The lowest unoccupied molecular orbital (LUMO) and highest occupied molecular orbital (HOMO) energy levels of the NaFSI, FSI⁻, DMTMSA, EMTMSA, DETMSA, and Na⁺-EMTMSA were calculated using the density functional theory (DFT) in Gaussian 09 W with B3LYP⁸⁻¹⁰/6-311+g (d, p)^{11,12} optimization.”

Comment 4: EMTMSA-based electrolyte demonstrates largely improved flame retardancy,

effectively suppressing ignition and sustaining non-flammability. Does this electrolyte have a negative impact on battery performance, such as capacity and coulombic efficiency? If there are negative impacts, how to strike a balance.

Response: We appreciate the reviewer for raising this important question regarding practical application. Our results clearly indicate that the EMTMSA-based electrolyte, while providing excellent flame retardancy, shows no observable negative impact on battery performance in terms of capacity and coulombic efficiency (CE). Specifically, the initial discharge capacity of Na||NFM cells (Fig. R8a and b, previous Supplementary Fig. 10c and f) with the EMTMSA-based electrolyte system (163.1 mAh g⁻¹ at 0.05 C) is superior to that of the carbonate-based electrolyte (161.5 mAh g⁻¹ at 0.05 C). This confirms that EMTMSA, as a major solvent, does not compromise Na⁺ transport kinetics or the initial utilization of active materials. Regarding CE (Fig. R8c), the EMTMSA-based electrolyte maintained an outstanding average CE of 99.94% after long-term cycling (1000 cycles), which is comparable to the 99.89% achieved by the carbonate-based electrolyte. Consequently, the EMTMSA-based electrolyte achieves a harmonious balance: its superior non-flammability greatly enhances battery safety, while its presence has no detrimental effect on electrochemical performance, and concurrently provides enhanced long-term operational stability.

Fig. R8 Electrochemical performance of cells with different electrolytes at room temperature. Voltage profiles of the Na||NFM cells with the (a) carbonate-based and (b) EMTMSA-based electrolytes at 4.2 V cut-off voltage at 0.05 C. (c) CEs of the NFM||HC pouch cells with different electrolytes at 4.2 V_{Na} cut-off voltage upon 1000 cycles.

Changes:

Main text:

“Compared to the carbonate-based electrolyte, the pouch cell with the EMTMSA-based electrolyte also delivers improved average Coulombic efficiency (CE) at 4.2 V_{Na} (~99.94%, Supplementary

Fig. 14) and better rate performance (Supplementary Fig. 15).”

Supplementary Information:

Updated Supplementary Fig. 14 Coulombic efficiencies of the pouch cells with different electrolytes at 4.2 V_{Na} cut-off voltage.

Comment 5: The safety of batteries is related to the self-generated heat initiation temperature T₁, the thermal runaway trigger temperature T₂, and the maximum thermal runaway temperature T₃. Whether the EMTMSA-based electrolyte can improve battery safety by changing T₁, T₂, and T₃? Please provide relevant evidence.

Response: We appreciate the reviewer’s comment on the cell-level safety, which is beyond the electrolyte level. In order to evaluate the cell-level safety, thermal runaway evaluation of NFM||HC pouch cells are conducted using an accelerating rate calorimetry (ARC, BTC130, HEL, England). The pouch cells are charged to 4.2 V at 0.1 C before test. During “heat-wait-search” mode, the heating step is set as 5 °C per step and the detection limit is 0.03 °C min⁻¹. The thermal runaway point is set as 1 °C min⁻¹.

As shown in Fig. R9, the EMTMSA-based electrolyte significantly elevates both the self-generated heat initiation temperature (T₁) and the thermal runaway trigger temperature (T₂), relative to the carbonate-based electrolyte, increasing T₁ from 163 °C to 206 °C and T₂ from 233 °C to 278 °C. This indicates that the thermal stability margin of the cell is effectively expanded. Simultaneously, the maximum thermal runaway temperature (T₃) of the EMTMSA-based electrolyte also showed a certain degree of reduction (decreasing from 471 °C to 468 °C). The increase in T₁ and T₂ is primarily attributed to the high flame retardancy of EMTMSA and the formation of a stable CEI, which inhibits vigorous exothermic side reactions between the electrolyte and the electrodes. The reduction in T₃ indicates that the severity of the thermal runaway is also mitigated. These ARC data verify that the flame-retardant properties of the EMTMSA molecule are effectively propagated to the pouch-cell level, highlighting the value of molecular-level flame-retardant electrolyte design.

Fig. R9 Temperature evolution monitored during the thermal runaway of fully charged pouch cells under ARC testing. Where T_1 represents the self-generated heat initiation temperature, T_2 represents the thermal runaway trigger temperature, and T_3 represents the maximum thermal runaway temperature. Compared to pouch cells with the carbonate-based electrolyte, these three key indicators are all reduced in pouch cells with EMTMSA-based electrolyte.

Changes:

Main text:

“In addition to improved high-temperature cycling stability, the EMTMSA-based electrolyte also delivers clear advantages under thermal-abuse conditions. Pouch cells charged to 100% state of charge (4.2 V) were evaluated by accelerated rate calorimetry (ARC) to track key thermal-runaway temperatures (Fig. 4f). The EMTMSA-based electrolyte markedly increases both the self-generated heat initiation temperature (T_1 , from 163 °C to 206 °C) and the thermal-runaway trigger temperature (T_2 , from 233 °C to 278 °C), while showing a slight reduction in the maximum thermal-runaway temperature (T_3 , from 471 °C to 468 °C). These ARC data verify that the flame-retardant properties of the EMTMSA molecule are effectively propagated to the pouch-cell level, highlighting the value of molecular-level flame-retardant electrolyte design.”

Updated Fig. 4 Electrochemical performance of 1 Ah NFM||HC pouch cells with different electrolytes at extreme conditions. Voltage profiles of pouch cells with the carbonate-based (a) and EMTMSA-based (b) electrolytes at 25 °C, -20 °C, -40 °C, -60 °C, and -70 °C. (c) Optical photographs of different electrolytes after overnight storage at -60 °C. The carbonate-based electrolyte was frozen while the EMTMSA-based electrolyte remained liquid. (d) Cycling performance of pouch cells with both charging and discharging at -20 °C. (e) Comparison of electrochemical performance between previous reports and this work, based on upper cut-off voltages and capacity retentions at a specific low temperature (detailed corresponding references are provided in Supplementary Table 3). The cell with the EMTMSA-based electrolyte in this study enables superior high-voltage operation and excellent capacity retention at low temperature outperforming previous arts. (f) Temperature evolution monitored during the thermal runaway of fully charged pouch cells under ARC testing. Where T_1 represents the self-generated heat initiation temperature, T_2 represents the thermal runaway trigger temperature, and T_3 represents the maximum thermal runaway temperature. Compared to pouch cells with the carbonate-based electrolyte, these three key indicators are all reduced in pouch cells with EMTMSA-based electrolyte.

Experimental section:

“Thermal runaway study of NFM||HC pouch cells are conducted using an accelerating rate calorimetry (ARC, BTC130, HEL, England). The pouch cells are charged to 4.2 V at 0.1 C before test. During “heat-wait-search” mode, the heating step is set as 5 °C per step and the detection limit is 0.03 °C min⁻¹. The thermal runaway point is set as 1 °C min⁻¹.”

Comment 6: In the battery cycle test depicted in Figure 3a and 3b, the airbag of the pouch cell is not removed. For the control group cell, gas is produced during the formation stage. If the gas is not completely removed, it will also have a negative impact on the battery cycle test, making it impossible to determine the effect of functional electrolyte components.

Response: We sincerely thank the reviewer for pointing out the effect of the gas pocket on the cycling performance.

During the formation cycle, gas generation is expected due to the CEI/SEI formation. To eliminate the influence of the gas on the cycling stability, pouch cells undergo a vacuum degassing followed by final sealing (Fig. R10a and b) after formation. In this process, the edge of the gas pocket is gently punctured to release the accumulated gas under vacuum, and the pouch is then hot-sealed under vacuum to ensure complete removal of residual gas (Fig. R10c, d). **This process effectively eliminates the influence of formation-induced gas generation on the subsequent cycling tests.**

A small gas pocket remains after final sealing (Fig. R10e), the pouch cells are placed in a mechanical fixture for cycling tests. In this configuration, any gas generated during the early cycles can be accommodated by the remaining gas pocket without affecting the stacking, thus minimizing its influence on cycling performance and enabling a more accurate evaluation of electrolyte–electrode interfacial stability. At the same time, the gas pocket serves as a visible indicator that allows qualitative observation of gas evolution during long-term cycling. Therefore, it can be expected that for the reference carbonate-based electrolyte, which generates more gas during cycling, the cycling performance would deteriorate even *earlier* if the gas pocket were completely removed.

Fig. R10 Final sealing process and the fixture for pouch cells. (a) Schematic illustration of the pouch cell final sealing and degassing processes before long-term cycling. (b) Photograph of the final sealing machine (equipped with a vacuum system and hot sealing function). (c) An enlarged photograph of the pouch cell under the hot blade of the final sealing machine. During sealing, the upper platen descends as the chamber is evacuated by a vacuum pump. Under vacuum, the edge of the gas pocket is gently punctured to release the accumulated gas, followed by hot-sealing of the residual pocket. This procedure ensures complete removal of trapped gas generated during the formation process. (d) Photograph of the pouch cell after the final sealing, with excess gas pocket removed. (e) Side-view (upper) and front-view (lower) photographs of the pouch cell mounted in the mechanical fixture after cycling. A visibly swollen gas pocket is observed along the cell edge, indicating significant gas generation during long-term cycling.

Changes:

Main text:

“The gas generated during the formation cycle was completely removed by a vacuum degassing process during the final sealing step after formation. A small gas pocket remains after final sealing, serving as a visible indicator for qualitative observation of gas evolution during long-term cycling, after which the pouch cells are placed in a mechanical fixture for cycling tests (Supplementary Fig. 10).”

Supplementary Information:

Updated Supplementary Fig. 10 Final sealing process and the fixture for pouch cells. (a) Schematic illustration of the pouch cell final sealing and degassing processes before long-term cycling. (b) Photograph of the final sealing machine (equipped with a vacuum system and hot sealing function). (c) An enlarged photograph of the pouch cell under the hot blade of the final sealing machine. During sealing, the upper platen descends as the chamber is evacuated by a vacuum pump. Under vacuum, the edge of the gas pocket is gently punctured to release the accumulated gas, followed by hot-sealing of the residual pocket. This procedure ensures complete removal of trapped gas generated during the formation process. (d) Photograph of the pouch cell after the final sealing, with excess gas pocket removed. (e) Side-view (upper) and front-view (lower) photographs of the pouch cell mounted in the mechanical fixture after cycling. A visibly swollen gas pocket is observed along the cell edge, indicating significant gas generation during long-term cycling.

Experimental section:

“Following electrolyte filling and hot sealing in the glove box, the pouch cells were pre-cycled at 40 °C at a low rate of 0.05 C for the formation step. After formation, the cells underwent a vacuum degassing followed by final sealing (Supplementary Fig. 10). In this process, the edge of the gas pocket is gently punctured to release the accumulated gas under vacuum, and the pouch is then hot-sealed under vacuum to ensure complete removal of residual gas. This process effectively eliminates the influence of formation-induced gas generation on the subsequent cycling tests. A small gas pocket remains after final sealing (Supplementary Fig. 10), the pouch cells are placed in a mechanical fixture for cycling tests.”

Comment 7: The use of EMTMSA-based electrolyte cell demonstrates excellent low-temperature performance, but the high-temperature performance of the cell is not mentioned in the article. How about the cycling performance at 45 °C and 60 °C?

Response: We appreciate the reviewer’s comments on high-temperature cycling stability of our electrolyte. We evaluated the electrochemical performance of pouch cells using both the carbonate-based and EMTMSA-based electrolytes at high temperatures (45 °C and 60 °C). As shown in Fig. R11, pouch cells with the EMTMSA-based electrolyte exhibit better cycling

stability at elevated temperatures compared to those with the carbonate-based electrolyte. Specifically, the capacity retention after 200 cycles at 45 °C is 75.7% for the carbonate-based electrolyte versus 87.7% for the EMTMSA-based electrolyte; at 60 °C, the retention is 71.4% and 82.1%, respectively. This result highlights the advantage of our electrolyte for wide-temperature operation.

Fig. R11 Cycling performance of pouch cells with the carbonate-based and EMTMSA-based electrolytes at high temperatures with an upper cut-off voltage of 4.2 V_{Na} at 0.5 C. (a) 45 °C. (b) 60 °C.

Changes:

Main text:

“Moreover, the high-temperature stability at 45 °C and 60 °C was evaluated in pouch cells at an upper charging voltage of 4.2 V_{Na} . Pouch cells with the EMTMSA-based electrolyte exhibit better cycling stability compared with those employing the carbonate-based electrolyte, achieving 87.7% capacity retention after 200 cycles at 45 °C (vs. 75.7%) and 82.1% retention at 60 °C (vs. 71.4%, Supplementary Fig. 20).”

Supplementary Information:

Updated Supplementary Fig. 20 Cycling performance of pouch cells with the carbonate-based and EMTMSA-based electrolytes at high temperatures with an upper cut-off voltage of $4.2 V_{Na}$ at $0.5 C$. (a) $45\text{ }^{\circ}C$. (b) $60\text{ }^{\circ}C$.

Comment 8: For electrolyte solvents, physicochemical parameters such as dielectric constant, melting and boiling points, flash point, and viscosity are crucial. Please provide the relevant parameters of EMTMSA to demonstrate its potential as a promising electrolyte solvent for sodium-ion batteries.

Response: We agree with the reviewer that the basic physicochemical parameters are important for electrolyte. The key physicochemical parameters of the EMTMSA solvent were summarized in Table R1. EMTMSA exhibits a low melting point ($-86\text{ }^{\circ}C$), a high flash point ($170\text{ }^{\circ}C$) and a high boiling point ($155\text{ }^{\circ}C$). The dielectric constant of EMTMSA is 8.21, which facilitates adequate dissolution of sodium salts, while its moderate viscosity (1.8 mPa s) ensures good fluidity and adequate ionic transport rates. The combination of these key physicochemical parameters strongly demonstrates the promising potential of EMTMSA as an electrolyte solvent for SIBs.

Table R1 Physicochemical parameters of the EMTMSA solvent.

	Melting point	Flash point	Boiling point	Dielectric constant	Viscosity
EMTMSA	$-86\text{ }^{\circ}C$	$170\text{ }^{\circ}C$	$155\text{ }^{\circ}C$	8.21	1.8 mPa s

Changes:

Main text:

“This molecular asymmetry disrupts solid-state packing, resulting in extremely poor crystallinity

and an ultralow melting point of $-86\text{ }^{\circ}\text{C}$ (Fig. 1d). In addition, its high flash and boiling points, along with moderate viscosity and dielectric constant, render it a well-balanced and practically viable electrolyte solvent (Supplementary Table 1).”

Supplementary information:

Updated Supplementary Table 1. Physicochemical parameters of the EMTMSA solvent.

	Melting point	Flash point	Boiling point	Dielectric constant	Viscosity
EMTMSA	$-86\text{ }^{\circ}\text{C}$	$170\text{ }^{\circ}\text{C}$	$155\text{ }^{\circ}\text{C}$	8.21	1.8 mPa s

Experimental section:

“The flash point was measured using a closed-cup flash point tester (SYD-3536-1). The dielectric constant was determined using a Wayne Kerr 6530B impedance analyzer. Viscosity was evaluated using an AMETEK Brookfield viscometer at room temperature. The boiling point of EMTMSA was obtained experimentally during synthesis.”

Comment 9: The author investigated the unique effects of several components, including DMTMSA, DETMSA, and EMTMSA, in different battery systems. Can these components also exhibit similar functional characteristics in the same battery material system?

Response: To verify whether DMTMSA, DETMSA, and EMTMSA exhibit similar functional characteristics within the same battery material system (the NFM||HC pouch cell), we have supplemented the cycling stability test results for the DMTMSA-based (Fig. R12a) and DETMSA-based (Fig. R12b) electrolytes (sulfonamide-based electrolytes were prepared by replacing EMTMSA with DMTMSA or DETMSA) under identical test conditions. Our test results show that the DMTMSA-based electrolyte retained 81.9% of its capacity after 1000 cycles (Fig. R12a). This highly similar performance to EMTMSA-based electrolyte and superior performance significantly outperforms the carbonate-based electrolyte (Fig. R12c). However, although pouch cell with the DMTMSA-based electrolyte exhibited good cycling stability at high voltage, its discharge capacity dropped to zero at $-60\text{ }^{\circ}\text{C}$ (Fig. R12d), a performance similar to, but slightly better than, that of the carbonate-based electrolyte at low temperatures (Fig. R12e), which limits its broader application scope. Conversely, although the DETMSA has a lower T_m ($-87\text{ }^{\circ}\text{C}$), indicating some potential for low-temperature applications, the DETMSA-based electrolyte exhibited noticeably poorer cycling stability under the same $4.2\text{ }V_{\text{Na}}$ high-voltage, retaining only 64.5% of its capacity after 1,000 cycles (Fig. R12b). Overall, only the EMTMSA-based electrolyte combines both high-voltage stability and low-temperature application potential (Fig. R12c, f).

Fig. R12 Long-term cycling performance and discharge capacity at low temperatures of the pouch cells with upper cut-off voltages of $4.2 V_{Na}$. Long-term cycling performance of the pouch cells with the (a) DMTMSA-based, (b) DETMSA-based, (c) carbonate-based and EMTMSA-based electrolytes. Discharge capacity at low temperatures of the pouch cells with the (d) DMTMSA-based, (e) carbonate-based, (f) EMTMSA-based electrolytes.

Changes:

Main text:

“More importantly, when evaluated at a higher cut-off voltage of $4.2 V_{Na}$ (Fig. 3b), the pouch cell with our EMTMSA-based electrolyte can still retain 81.6% of its initial capacity after 1,000 cycles at charging-discharging rate of 0.5 C (comparable with the pouch cell with the DMTMSA-based electrolyte, Supplementary Fig. 12a), outperforming the carbonate-based and DETMSA-based electrolytes (Supplementary Fig. 12b), which retain 65.6% and 64.5% of its original capacity for 1000 cycles under the same conditions, respectively.”

“Similarly, the capacity of the pouch cell using the DMTMSA-based electrolyte declines to nearly zero at $-60\text{ }^{\circ}\text{C}$ (Supplementary Fig. 17).”

Supplementary Information:

Updated Supplementary Fig. 12 Long-term cycling performance of the pouch cells with upper cut-off voltages of 4.2 V_{Na}. (a) DMTMSA-based electrolyte. (b) DETMSA-based electrolyte.

Supplementary Fig. 17 Voltage profiles of the pouch cell with the DMTMSA-based electrolyte during discharging at 25 °C, -20 °C, -40 °C, and -60 °C.

Reviewer #3 (Remarks to the Author):

This manuscript introduces a novel non-flammable, low-melting-point sulfonamide molecule, EMTMSA, as a solvent for sodium-ion battery electrolytes. The system demonstrates fast sodium-ion transport and stable electrode–electrolyte interphase formation. The authors conducted extensive material characterizations and evaluated electrochemical performance in Ah-level pouch cells, which showed promising high-voltage and low-temperature performance with potential for practical applications. However, several conclusions and discussions require clarification or further justification. I therefore recommend a major revision before this manuscript can be considered for publication. Specific comments are as follows:

Response: We appreciate the reviewer’s positive comments on our work.

Comment 1: What is the relative percentage of SSIP, CIP, and AGG in both EMTMSA- and carbonate-based electrolytes? The manuscript states that AGG and CIP dominate in the EMTMSA-based electrolyte, yet the solvation structure in Fig. 2f does not show any FSI⁻ anions.

Response: We thank the reviewer for of the comments on electrolyte solvation structure. The percentages of SSIP, CIP, and AGG were quantified by fitting the peak areas of the Raman spectra (Fig. R13a and b). While SSIP (56.4%) and CIP (43.6%) dominate the solvation environment in the carbonate-based electrolyte, the EMTMSA-based electrolyte is instead characterized by a much higher proportion of CIP (84.2%) and AGG (10.2%). The moderate polarity of the EMTMSA solvent facilitates partial dissociation of Na⁺ and FSI⁻, allowing a portion of FSI⁻ to enter the first solvation sheath. We also adjust the schematic illustration of the solvation structure of the EMTMSA-based electrolyte by including FSI⁻ anion (Fig. R13c, d). Fig. R13 has been updated in the revised manuscript as updated Fig. 2, which has been highlighted in yellow.

Fig. R13 Deconvolution of Na⁺–FSI⁻ Raman peaks for the (a) carbonate-based and (b) EMTMSA-based electrolytes with peak fits for different cluster types including SSIP, CIP, and AGG. Schematic figures of the representative Na⁺ solvation structures in the (c) carbonate-based and (d) EMTMSA-

based electrolytes.

Changes:

Main text:

“While SSIP (56.4%) is the dominant cluster type in the carbonate-based electrolyte (Fig. 2a), AGG (10.2%) and CIP (84.2%) are the main species in the EMTMSA-based electrolyte (Fig. 2b), indicating that more FSI⁻ anions participate in the first solvation sheath.”

Updated Fig. 2 The coordination interactions between Na⁺ and different solvent molecules in the EMTMSA-based and carbonate-based electrolytes by Raman spectroscopy and NMR techniques. Deconvolution of Na⁺-FSI⁻ Raman peaks for the carbonate-based (a) and EMTMSA-based (b) electrolytes with peak fits for different cluster types including SSIP, CIP, and AGG. ¹⁷O NMR spectra of the carbonate-based (c) and EMTMSA-based (d) electrolytes before and after incorporating 0.1 M and 1 M NaFSI into the solvent mixtures. The corresponding functional groups in different molecules and chemical shifts are denoted. Schematic figures of the representative Na⁺ solvation structures in the carbonate-based (e) and EMTMSA-based (f) electrolytes. ¹H VT-NMR spectra of the PC/EMC (g) and EMTMSA/PC/EMC (h) solvent mixtures at 25 °C, -20 °C, and -50 °C. Peak broadening is observed for the carbonate mixtures upon cooling, corresponding to significantly decreased molecular rotation. In contrast, the peaks remain almost unchanged for the EMTMSA-carbonate mixture.

Comment 2: For low-temperature operation of SIBs, charging is usually much more difficult than discharging. Since Figs. 4a and 4b only display LT discharge capacity, can the cells also be charged under the same conditions? Long-term viability at extreme temperatures should also be demonstrated.

Response: We agree with the reviewer that it is challenging for the cell to charge at the low temperatures. To address this comment, we performed more tests at low temperatures as below.

1. Fig. R14a and b show the voltage profiles for pouch cells using the carbonate-based and

EMTMSA-based electrolytes, respectively, *under identical low-temperature ($-20\text{ }^{\circ}\text{C}$, $-40\text{ }^{\circ}\text{C}$, and $-60\text{ }^{\circ}\text{C}$) charge-discharge conditions*. The EMTMSA-based electrolyte largely outperforms the carbonate-based one at low temperatures. Whereas the carbonate electrolyte retains 60.3% of its room-temperature capacity at $-20\text{ }^{\circ}\text{C}$, drops sharply to 8.1% at $-40\text{ }^{\circ}\text{C}$, and becomes inactive at $-60\text{ }^{\circ}\text{C}$, the EMTMSA-based electrolyte maintains 81.1% at $-20\text{ }^{\circ}\text{C}$, 59.7% at $-40\text{ }^{\circ}\text{C}$, and still delivers an impressive 41.9% retention even at $-60\text{ }^{\circ}\text{C}$.

- The EMTMSA-based electrolyte also outperforms the carbonate-based system in cycling stability when both charging and discharging are carried out at $-20\text{ }^{\circ}\text{C}$, as demonstrated by the long-term cycling performance (Fig. R15). Under these conditions, the EMTMSA-based cell retains 59.2% of its initial capacity after 500 cycles, whereas the carbonate-based cell maintains only 37.0%.

Fig. R14 Voltage profiles of pouch cells with the (a) carbonate-based and (b) EMTMSA-based electrolytes during charge/discharge at a rate of 0.01 C, measured at $25\text{ }^{\circ}\text{C}$, $-20\text{ }^{\circ}\text{C}$, $-40\text{ }^{\circ}\text{C}$, and $-60\text{ }^{\circ}\text{C}$.

Fig. R15 Long-term electrochemical performance of 1 Ah NFM||HC pouch cells with different electrolytes at $-20\text{ }^{\circ}\text{C}$, with both charging and discharging at 0.1 C.

Changes:

Main text:

“The pouch cell with the EMTMSA-based electrolyte maintained excellent capacity retention, even under simultaneous low-temperature charging and discharging, retaining 81.1%, 59.7%, and 41.9% of its room temperature capacity at $-20\text{ }^{\circ}\text{C}$, $-40\text{ }^{\circ}\text{C}$, and $-60\text{ }^{\circ}\text{C}$, respectively (Supplementary Fig. 18).”

“Compared to the carbonate-based electrolyte, the pouch cell with the EMTMSA-based electrolyte delivers improved cycling stability and CE over 500 cycles with both charging and discharging at $-20\text{ }^{\circ}\text{C}$ (Fig. 4d; Supplementary Fig. 19).”

Updated Fig. 4 Electrochemical performance of 1 Ah NFM||HC pouch cells with different electrolytes at extreme conditions. Voltage profiles of pouch cells with the carbonate-based (a) and EMTMSA-based (b) electrolytes at $25\text{ }^{\circ}\text{C}$, $-20\text{ }^{\circ}\text{C}$, $-40\text{ }^{\circ}\text{C}$, $-60\text{ }^{\circ}\text{C}$, and $-70\text{ }^{\circ}\text{C}$. (c) Optical photographs of different electrolytes after overnight storage at $-60\text{ }^{\circ}\text{C}$. The carbonate-based electrolyte was frozen while the EMTMSA-based electrolyte remained liquid. (d) Cycling performance of pouch cells with both charging and discharging at $-20\text{ }^{\circ}\text{C}$. (e) Comparison of electrochemical performance between previous reports and this work, based on upper cut-off voltages and capacity retentions at a specific low temperature (detailed corresponding references are provided in Supplementary Table 3). The cell with the EMTMSA-based electrolyte in this study enables superior high-voltage operation and excellent capacity retention at low temperature outperforming previous arts. (f) Temperature evolution monitored during the thermal runaway of fully charged pouch cells under ARC testing. Where T_1 represents the self-generated heat initiation temperature, T_2 represents the thermal runaway trigger temperature, and T_3 represents the maximum thermal runaway temperature. Compared to pouch cells with the carbonate-based electrolyte, these three key indicators are all reduced in pouch cells with EMTMSA-based electrolyte.

Supplementary Information:

Updated Supplementary Fig. 18 Voltage profiles of pouch cells with the (a) carbonate-based and (b) EMTMSA-based electrolytes during charge/discharge at a rate of 0.01 C, measured at 25 °C, -20 °C, -40 °C, and -60 °C.

Comment 3: The authors mentioned that the suppressed Na precipitation observed in the EMTMSA-based electrolyte can be attributed to its lower cell impedance at low temperatures. However, the evolution in cell impedance at LT is not provided.

Response: We agree with the reviewer that the impedance evolution of the cells at low temperature is important to support our argument. *In situ* EIS-DRT analysis for the pouch cells using both the EMTMSA-based and carbonate-based electrolytes at -20 °C was performed (Fig. R16). At -20 °C, the EMTMSA-based cell exhibits much lower R_e (electrolyte resistance) and $R_{SEI/CEI}$ (SEI and CEI resistance) compared to the carbonate electrolyte during the entire charge and discharge process. This indicates that the EMTMSA-based electrolyte enables the formation of interphases with faster Na^+ transport and facilitates more efficient Na^+ desolvation at low temperatures.

Fig. R16 *In situ* distribution of relaxation times data of the pouch cells with carbonate-based (left) and EMTMSA-based (right) electrolytes at -20 °C upon charging and discharging between 1.5 V to 4.2 V.

Changes:

Main text:

“The suppressed Na^0 precipitation observed in the EMTMSA-based electrolyte can be attributed to its lower cell impedance at low temperatures (Supplementary Fig. 22), as well as the formation of a more ionic conductive and stable solid–electrolyte interphase (SEI) on the HC anode.”

Supplementary Information:

Updated Supplementary Fig. 22 *In situ* distribution of relaxation times data of the pouch cells with carbonate-based (left) and EMTMSA-based (right) electrolytes at $-20\text{ }^{\circ}\text{C}$ upon charging and discharging between 1.5 V to 4.2 V. At $-20\text{ }^{\circ}\text{C}$, the EMTMSA-based cell exhibits much lower R_e (electrolyte resistance) and $R_{\text{SEI/CEI}}$ (SEI and CEI resistance) compared to the carbonate electrolyte during the entire charge and discharge process.

Comment 4: How to interpret the “negative” capacity losses associated with EMTMSA electrolyte in Fig. 5f.

Response: We thank the reviewer for their attention to the results of the capacity loss test following various pause times in Fig. 5f. For the carbonate-based electrolyte, a partial positive capacity loss was observed after experiencing different pause times. This clearly indicates that the corresponding SEI underwent partial dissolution and reorganization during the rest period, suggesting its poor ability to suppress side reactions. In contrast, in the EMTMSA-based electrolyte, we observed “negative capacity losses”. This confirms that no SEI dissolution occurred; rather, the capacity change reflects a natural capacity variation following the rest period. This clearly reflects that the lower polarity of the EMTMSA-based electrolyte helps preserve SEI integrity and suppress excessive thickening. Notably, this “negative capacity losses” phenomenon has also been reported in *Adv. Mater.* e15562 (2025).

Changes:

Main text:

“In contrast, the lower polarity of the EMTMSA-based electrolyte helps preserve SEI integrity and suppress excessive thickening⁴⁸.”

Reference:

“48. Xu, L. *et al.* 4.5-V-class safe lithium-ion batteries with silicon-majority-graphite anodes enabled by self-limiting interphase. *Adv. Mater.* e15562 (2025).”

Comment 5: SEI formed on hard carbon and Cu substrates can differ in chemical composition and thickness. It is recommended to include SEI characterization of hard carbon anodes after cycling.

Response: We thank the reviewer for this suggestion. We have conducted surface chemistry analysis on the cycled HC anodes by XPS (Fig. R17). The XPS C1s and O1s data reveal that the SEI formed in the carbonate-based electrolyte contains more organic species (C ratio: 37.4%,

such as C–O and C=O), while it is significantly reduced in the EMTMSA-based electrolyte, lowering to 25.2% (Fig. R17a and b, calculated based on the ratio of C element associated signals to all detected elements). The C1s data of the SEI formed in the carbonate-based electrolyte shows a Na_xC peak (Fig. R17a), corresponding to the formation of Na precipitation observed in the SEM images of the HC anode extracted from the cycled pouch cell. Furthermore, the XPS F1s data shows more NaF species are present in the EMTMSA-derived SEI (Fig. R17c). These results demonstrate that the SEI formed by the EMTMSA-based electrolyte is more inorganic with NaF-rich components, consistent with our TOF-SIMS data obtained from the Cu substrate.

Fig. R17 has been updated as Fig. 5g and Supplementary Fig. 23, and related discussion has been incorporated, which are both highlighted in yellow in the revision.

Fig. R17 XPS analysis for the SEIs formed on HC with different electrolytes. (a) C1s spectra of the SEIs formed on HC. (b) O1s spectra of the SEIs formed on HC. (c) F1s spectra of the SEIs formed on HC.

Changes:

Main text:

“X-ray photoelectron spectroscopy (XPS) results show that the SEI formed on HC in the EMTMSA-based electrolyte contains a higher proportion of NaF species and less organic species without detectable Na_xC (corresponding to the detrimental sodium precipitation), compared to that formed in the carbonate-based electrolyte (Fig. 5g, Supplementary Fig. 23).”

Updated Fig. 5 Characterizations of SEIs formed on HC anodes with different electrolytes. Optical (a) and SEM and EDS elemental mapping (b) images of the HC anodes extracted from the pouch cells in different electrolytes at $-20\text{ }^{\circ}\text{C}$ for 200 cycles. Obvious Na^0 precipitation was observed from the one cycled in the carbonate-based electrolyte (b). Scale bars 10 μm (a) and 5 μm (b). High-resolution TEM images of the HC particles cycled in the carbonate-based (c) and EMTMSA-based (d) electrolytes. Scale bars, 10 nm. (e) Evolution of the capacity contributed by electrolyte reduction before and after each pause as a function of cycle number by $\text{Na}||\text{Cu}$ cells with different electrolytes. The grey-shaded regions with arbitrary width indicate the periods during which the cells were paused. (f) Capacity loss associated with different pause durations for cells using carbonate-based and EMTMSA-based electrolytes. Each shaded curve represents an individual cell, while the solid lines track the average values from three parallel cells. (g) F 1s XPS spectra of the SEIs formed in both electrolytes on HC. TOF-SIMS depth profiles and 3D reconstructed images of the interested species NaF_2^- , CHO_2^- , and Cu^- signals for SEIs formed in the carbonate-based (h) and EMTMSA-based (i) electrolytes. Blue and red colors correspond to the minimum and maximum intensities respectively.

Supplementary Information:

Updated Supplementary Fig. 23 XPS analysis for the SEIs formed on HC with different electrolytes. (a) C 1s spectra of the SEIs formed on HC. (b) O 1s spectra of the SEIs formed on HC.

Comment 6: What sputtering rate was used in TOF-SIMS? Some experimental details are missing in the Supporting Information and should be clarified.

Response: The size of the sputtering beam and sputtering rate is $250 \times 250 \mu\text{m}^2$ and 0.1 nm s^{-1} , respectively. These parameters have been included in the experimental section, which has been highlighted in yellow.

Changes:

Experimental Section:

“Surface chemistry of the cycled electrodes was analyzed by X-ray photoelectron spectroscopy (XPS, Thermo Fisher Escalab Xi⁺) and time of flight secondary ion mass spectrometer (TOF-SIMS, ION-TOF M6, sputtering beam $250 \times 250 \mu\text{m}^2$, sputtering rate 0.1 nm s^{-1}).”

Comment 7: Given the structural similarity between NaFSI and EMTMSA, how can one distinguish whether the inorganic SEI/CEI components originate from salt or EMTMSA decomposition? Does this imply that EMTMSA also contributes to interfacial stability via sacrificial decomposition? A DFT analysis of HOMO–LUMO levels may help identify decomposition preference.

Response: We highly appreciate this comment regarding the decomposition contributions of the NaFSI salt and EMTMSA solvent. Following the reviewer’s suggestion, we calculated the LUMO and HOMO energy levels by DFT calculations for the FSI⁻, NaFSI, EMTMSA, and Na⁺–EMTMSA (Fig. R18).

Anodic stability: bare EMTMSA exhibits the highest HOMO level (-7.8983 eV), indicating it is intrinsically the easiest species to oxidize. However, upon coordination with Na⁺, its HOMO is greatly lowered to -11.6280 eV . This substantial stabilization suggests that Na⁺–EMTMSA becomes much more resistant to oxidation and is therefore more anodically stable than the FSI⁻

anion (-10.4902 eV). This trend implies that, under such solvation conditions, oxidative decomposition would preferentially occur on FSI⁻ rather than on Na⁺-EMTMSA, consistent with the formation of F-containing CEI species and the enhanced oxidative stability of the EMTMSA-based electrolyte.

Cathodic stability: the LUMO levels follow the order EMTMSA (-0.3391 eV) > NaFSI (-2.1706 eV) > FSI⁻ (-2.4865 eV) > Na⁺-EMTMSA (-5.2319 eV). The relatively high LUMO of bare EMTMSA suggests that the bare solvent molecule is intrinsically the most reduction-stable species. Coordination with Na⁺ markedly lowers its LUMO, making the Na⁺-EMTMSA complex substantially more prone to reduction than FSI⁻. This trend implies that, under such solvation conditions, reductive decomposition would preferentially occur on Na⁺-EMTMSA, consistent with the inorganic-rich SEI/CEI composition.

Fig. R18 has been updated as Supplementary Fig. 25, and related discussion has been incorporated, which are both highlighted in yellow in the revision.

Fig. R18 The LUMO and HOMO energy levels of the FSI⁻, NaFSI, EMTMSA, and Na⁺-EMTMSA calculated by DFT calculations. The trends in energy levels imply that under such solvation conditions, oxidative decomposition preferentially occurs on FSI⁻ while reductive decomposition is more likely on Na⁺-EMTMSA, collectively leading to the formation of inorganic- and F-containing interphases in the EMTMSA-based electrolyte.

Changes:

Main text:

“This reflects the advantage of our EMTMSA molecule with a sulfonimide-salt-like molecular structure that can deduce similar decomposition products as the sulfonimide salt⁵¹ to effectively passivate the high-voltage aggressive cathode surface (Supplementary Fig. 25).”

Supplementary Information:

Updated Supplementary Fig. 25 The LUMO and HOMO energy levels of the FSI⁻, NaFSI, EMTMSA, and Na⁺-EMTMSA calculated by DFT calculations. The trends in energy levels imply that under such solvation conditions, oxidative decomposition preferentially occurs on FSI⁻ while reductive decomposition is more likely on Na⁺-EMTMSA, collectively leading to the formation of inorganic- and F-containing interphases in the EMTMSA-based electrolyte.

Experimental section:

“The lowest unoccupied molecular orbital (LUMO) and highest occupied molecular orbital (HOMO) energy levels of the FSI⁻, NaFSI, DMTMSA, EMTMSA, DETMSA, and Na⁺-EMTMSA were calculated using the density functional theory (DFT) in Gaussian 09 W with B3LYP⁸⁻¹⁰/6-311+g (d, p)^{11,12} optimization).”

Reviewer #4 (Remarks to the Author):

Response: We sincerely appreciate your contribution as a co-reviewer of our manuscript. Your time and expertise have greatly contributed to strengthening our work, and we are grateful for your insights alongside those of the primary reviewer.